# Mathematical modeling of COVID-19 transmission dynamics in Uganda: Implications of complacency and early easing of lockdown

**Joseph Y. T. Mugisha[1]**☯, **Joseph Ssebuliba[1]**, **Juliet N. Nakakawa[1]**, **Cliff R. Kikawa[2]**,
**Amos Ssematimba**[3]☯*

**1** Department of Mathematics, College of Natural Sciences, Makerere University, Kampala, Uganda,
**2** Department of Economics and Statistics, Faculty of Economics and Management Sciences, Kabale
University, Kabale, Uganda, **3** Department of Mathematics, Faculty of Science, Gulu University, Gulu,
Uganda

☯ These authors contributed equally to this work.
* amos.ssematimba@gmail.com

University, SINGAPORE

**Data Availability Statement:** All relevant data are
within the manuscript and its Supporting
Information files.

## Abstract

### Background

Uganda has a unique set up comprised of resource-constrained economy, social-economic
challenges, politically diverse regional neighborhood and home to long-standing refuge cri-
sis that comes from long and protracted conflicts of the great lakes. The devastation of the
on-going global pandemic outbreak of severe acute respiratory syndrome coronavirus 2
(SARS-CoV-2) is likely to be escalated by these circumstances with expectations of the
impact of the disease being severe.

### Materials and methods

In this study, we formulate a mathematical model that incorporates the currently known dis-
ease characteristics and tracks various intervention measures that the government of
Uganda has implemented since the reporting of the first case in March 2020. We then evalu-
ate these measures to understand levels of responsiveness and adherence to standard
operating procedures and quantify their impact on the disease burden. Novel in this model
was the unique aspect of modeling the trace-and-isolate protocol in which some of the
latently infected individuals tested positive while in strict isolation centers thereby reducing
their infectious period.

### Results

The study findings show that even with elimination of all imported cases at any given time it
would take up to nine months to rid Uganda of the disease. The findings also show that the
optimal timing of easing of lockdowns while mitigating the possibility of re-emergence of a
second epidemic wave requires avoiding the scenario of releasing too-many-too-soon. It is

**Funding:** JYTM was funded through Makerere University Research and Innovation Fund.

**Competing interests:** The authors have declared that no competing interests exist.

even more worrying that enhancing contact tracing would only affect the magnitude and timing of the second wave but cannot prevent it altogether.

## Conclusion

We conclude that, given the prevailing circumstances, a phased-out lifting of lockdown measures, minimization of COVID-19 transmissibility within hospital settings, elimination of recruitment of infected individuals as well as enhanced contact tracing would be key to preventing overwhelming of the healthcare system that would come with dire consequences.

## 1. Introduction

A pandemic involving a fast spreading and fatal COVID-19, caused by severe acute respiratory syndrome coronavirus 2 (SARS-CoV-2) is currently impacting livelihoods globally [1, 2]. Its impact on global public health and economies emanates directly from the disease and the unintended consequences of the intervention measures implemented to curb its spread. It is associated with high morbidity and mortality levels and neither has a known cure nor a vaccine. Although still ongoing, its burden has already exceeded those of associated coronavirus namely, severe acute respiratory syndrome and Middle East respiratory syndrome [3].

On 21st March 2020, Uganda reported her first case of COVID-19 which was imported [4]. Subsequently, more predominantly imported cases were detected and the government responded by issuing the following guidelines to curb the spread—closure of schools, religious places, non–essential businesses, entertainment places, the stoppage of public transport, the social distancing and stay home–stay safe campaigns. The measures were intended to serve the purpose of; eliminating crowding places that would provide breeding grounds of new cases, limiting introduction of imported cases, limiting the chances of community transmission, and preventing the potential long-distance pathogen spread through vehicle facilitated movement. It is estimated that these measures reduced the susceptible population to about 10% initially (comprising mainly of essential workers and the non-complaint), with the fraction increasing to around 30% upon partial easing within the first two months.

These measures greatly slowed down the spread of the disease and accorded Uganda precious time to prepare and mobilize resources. Due to climatic, demographic and other differences between populations as well as country-level variations in the timing [5] and the nature of interventions, it is a challenge to come up with a one-fits-all approach to dealing with novel outbreaks globally. Consequently, designing efficient and effective region-specific intervention measures is key. This design process can be guided by theoretical studies involving mathematical models that incorporate the deployed interventions [5–12] as so far demonstrated for several countries and the World Health Organization [5, 8, 10, 13–17]. Such models provide a platform to perform scenario analyses through which in-depth insights into epidemic trends can be gained and the impact of the proposed interventions on disease transmission can be assessed. However, modeling the transmission dynamics of COVID-19 which is a novel and highly transmissible disease that can also be spread by asymptomatic individuals with a long incubation period is a challenge [18].

Regional differences call for the development of elaborate and locally parameterized mathematical models for more reliable predictions. The role of compartmental mathematical models in predicting COVID-19 dynamics under varying interventions has been demonstrated. Examples of such studies include, Yang et al. [19] who effectively predicted the epidemic peaks

and sizes in China using an approach that integrated population migration and COVID-19 epidemiological data into a Susceptible-Exposed-Infectious-Removed (SEIR) model together with an artificial intelligence approach. Also, Carcione et al. [20] demonstrated how mathematical models can be used to generate meaningful insights amidst epidemiological and etiological data challenges. They simulated a COVID-19 epidemic for Italy based on an SEIR model and they emphasized the importance of assessing the effectiveness of lockdowns on epidemic burden.

Relatedly, Chintalapudi et al. [21] concluded basing on outcomes of their data driven model analysis that if the Italian government extended the March 2020 country lockdown and self-isolation measures by 60 days, the number of registered cases would decrease by 35% while those recovered would increase by 66%. All these studies demonstrate the utility of mathematical models in assessing the impact of intervention measures on COVID-19 dynamics. On using modeling to assess how efficiently lockdown restrictions can be eased in the United Kingdom, Rawson et al. [22] applied an optimal control framework to an adapted SEIR model framework and found that the optimal strategy was a phased-out release.

Country-variations in the implementation of COVID-19 control measures among others may require formulation and parametrization of country-specific mathematical models. It is noteworthy that the government of Uganda initially implemented one of the strictest control measures in Africa that, among others, included institutional isolation of all arriving individuals, contact persons of confirmed cases and a strictly enforced lockdown of all non-essential activities [23].

We developed a mathematical model to study the dynamics of COVID-19 in Uganda and use it to forecast the disease trend under various intervention scenarios. The model was uniquely formulated in a such a way that it captures the trace-and-isolate protocol that was strictly implemented in Uganda. Additionally, although there are other approaches to modeling lockdown (e.g. [22]), our approach of having a given percentage of the susceptible population being totally unavailable to mingle was motivated by Uganda's strictness in enforcing lockdown measures. Using this model, implications of complacency, non-adherence to the standard operating procedures (SOPs) to curb disease spread as well as the timing and magnitude of easing lockdown are assessed. Ultimately, the study generates information to guide to the design of intervention measures suitable for the country and by extension, for other countries with similar demographic and other generic characteristics.

## 2. Materials and methods

This study uses data and information that have been reported by the World Health Organization (WHO) along with data reported by the Ministry of Health (MoH) Uganda during press briefings since the start of the pandemic. Data from existing literature on COVID-19 were also used in the parametrization process.

### 2.1 Model formulation, approach and variables

We develop a deterministic ordinary differential equation (ode) based epidemic model to capture the transmission dynamics of the disease. The model assumes frequency-dependent disease transmission and a homogeneous-mixing approximation. The homogeneous-mixing approximation—translating into the fact that every individual in the population having an equal chance of interacting with any other individual)- is motivated by the observed transmission patterns in which there seems to be no geographic clusters whereas the frequency-dependence in the transmission term is deemed suitable since the population size is large. Moreover, if the population size remains more or less constant as an epidemic passes through, then

density- and frequency-dependent models are equivalent [24–29]. The underlying compartmental model is of the *SEIHR(S)* format with the population split into compartments determined by their disease status. These include $S(t)$ representing the susceptible population $E(t)$ representing the latently infected individuals, the $I(t)$-class split into $I_a(t)$ for the asymptomatic infectious and $I_s(t)$ for the symptomatic infectious individuals, $H(t)$ for the hospitalized cases, and $R(t)$ representing the recovered individuals.

**2.1.1 Transitions through compartments.** In the model, individuals move between the compartments at rates determined by the underlying processes. These processes include infection of an $S(t)$ individual who consequently becomes latently infected. Upon completion of the latent period, a fraction ($c$) of the $E(t)$ individuals proceeds to become infectious while already in isolation in institutional quarantines. Of the remaining untraced $E(t)$ individuals, a fraction ($r$) proceeds to become asymptomatic while the rest become symptomatically infectious. The infectious individuals that remain in the community are subsequently identified and hospitalized (into $H(t)$ class) at rates $\omega_a$ and $\omega_s$ respectively. The disease can be fatal and symptomatic $I(t)$ individuals can succumb to the disease prior to hospitalization at a per capita rate ($\sigma_s$) or at a reduced rate ($\sigma_h$) upon hospitalization. $H(t)$ individuals proceed to recover (into $R(t)$ class) at a per capita rate ($\alpha$) where they are assumed to be (temporarily) immune to the disease. The model incorporates recruitment of susceptible, latently infected (a fraction ($e$)) and asymptomatically infectious (a fraction ($a$)) individuals and departures of susceptible or recovered individuals at a per capita rate $\lambda$.

**2.1.2 Model assumptions.** In addition to the generic assumptions on mixing pattern and transmission term formulation stated above, the other assumptions made include:

a. For the relatively short-term dynamics, an epidemic model in which the vital dynamics (i.e., birth and natural mortality) are ignored is opted for.

b. Upon infection, some latently infected individuals can be identified (e.g., through contact tracing) and subsequently individually isolated under high biosecurity conditions leading to their eventual hospitalization upon testing positive (i.e. completing their latent period), thereby denying them a chance of ever infecting other individuals in the community, while others may remain in the community and become infectious up until when they are identified and hospitalized.

c. In relation to the disease transmission potential, like the hospitalized individuals, individuals in institutional quarantine such as the traced contacts that later turn out to have been latently infected at the moment of their quarantine are assumed not to participate in disease transmission while in supervised quarantine centers.

d. Asymptomatically infectious individuals are assumed to be less likely to succumb to the disease but this is still being debated globally, with initial studies indicating cases of lung damage development in such individuals [30].

e. On disease recovery, the model parameter ($\tau$) can be varied to capture situations of varying duration of disease-induced immunity, ranging from no immunity to temporary and to life-long immunity at simulation level.

f. The arriving individuals (i.e., truck drivers, returnees and refugees etc.) comprise of susceptible, latently infected and asymptomatic infectious individuals while those exiting are susceptible and perhaps (later) the recovered.

g. The hospitalized individuals have potential of spreading the disease (i.e., hospital-acquired infections) albeit at a lower rate than the free-leaving infectious individuals.

h. Although viral loads have been reported to be similar between asymptomatic and symptomatic patients [31], the asymptomatically infected individuals may have a reduced infectivity because they may not cough or sneeze as much as the symptomatic. This however needs to be further elucidated.

i. The analysis was performed as the pandemic was still ongoing in Uganda, and as such the first 58 days of the reported confirmed cases were used to examine the fit of the model. Consequently, day 58 was crucial in the initiation of most of the simulated scenarios.

## 2.2 Model parameters

The disease-specific parameters were derived from existing relevant literature and contextualized to the region of interest. For the parameters where there was possible uncertainty, a plausible range was set and a sensitivity analysis performed. To capture the potential impact of the variation in intervention measures over time, the new recruitments were modeled by time-dependent step functions to mimic the variations in implemented measures. Following the approaches in [10, 18], the parameters (namely, $a$ and $e$) that would require further research for refinement, values within biologically feasible and practical ranges were chosen so that the simulations aligned with the reported cases in the validation step. Table 1 presents the description and values of parameters used in this study. For some of the parameter values derived from the Ministry of Health press briefs and other government agency websites, we describe what information was utilized in their respective footnotes in Table 1.

## 2.3 Conceptual model and equations

Based on the above descriptions, the corresponding compartmental model to capture the transmission dynamics of the disease is presented in Fig 1. Based on the details given above, we obtain the following system of odes (Eqs 3.1–3.6) to capture the transmission dynamics of the disease in Uganda:

$$\frac{dS}{dt} = (1 - (a + e))\lambda N - \frac{\beta b S(q I_a + I_s + gH)}{N} + \tau R - \lambda S \tag{3.1}$$

$$\frac{dE}{dt} = e\lambda N + \frac{\beta b S(q I_a + I_s + gH)}{N} - \rho E \tag{3.2}$$

$$\frac{dI_a}{dt} = a\lambda N + r\rho E - \omega_a I_a \tag{3.3}$$

$$\frac{dI_s}{dt} = (1 - (c + r))\rho E - \sigma_s I_s - \omega_s I_s \tag{3.4}$$

$$\frac{dH}{dt} = c\rho E + \omega_a I_a + \omega_s I_s - \sigma_h H - \alpha H \tag{3.5}$$

$$\frac{dR}{dt} = \alpha H - \tau R - \lambda R \tag{3.6}$$

where $N(t) = S(t) + E(t) + I_a(t) + I_s(t) + H + R(t)$.

In Eq (3.1), the term $(1-(a+e))\lambda N$ represents the number of arriving susceptible individuals per day, $\frac{\beta b S(q I_a + I_s + gH)}{N}$ represents the number of susceptible individuals that become latently

**Table 1. Summary of the parameters used in the disease transmission model as obtained from WHO and MoH (Uganda) briefings and reports as well as from relevant literature as indicated.**

| Parameter | Description | Value/range (units) | Source(s) |
|---|---|---|---|
| $a(t)$ | Percentage of arrivals that are asymptomatic | 0.000015–0.018 | Calibrated |
| $e(t)$ | Percentage of arrivals that are latently infected | 0.000015–0.018 | Calibrated |
| $\lambda$ | Per capita recruitment rate | 1.5/43000 (per day) | [42][a] |
| $\beta$ | Disease transmission rate | 0.5944 (per day) | [9] |
| $b$ | Percentage of susceptible individuals that is available | 10% | Estimated based on UBOS data [43] and Covid-19 control measures [23] [b] |
| $g$ | Infectivity factor for hospitalized individuals | 0.01 | Assumed based on [44][c] |
| $q$ | Infectivity factor among asymptomatic individuals | 1 | Assumed |
| $\alpha$ | Recovery rate of hospitalized individuals | 0.05 (per day) | Estimated based on [4, 45] |
| $\tau$ | Waning rate of disease-induced immunity | 0* (per day) | Assumed |
| $\rho$ | Progression rate from latent stage to infectious stage | 0.2 (per day) | [17] |
| $r$ | Percentage of latently infected individuals in community that becomes asymptomatic | 56% | [40, 41] |
| $\sigma_s$ | Disease induced mortality rate in symptomatic non-hospitalized individuals | 0.008 | Estimated based on [4, 46] |
| $\sigma_h$ | Disease-induced mortality rate for hospitalized individuals | 0.0008 | Assumed based on [44] |
| $\omega_a$ | Hospitalization rate of asymptomatic infectious | 0.2 (per day) | Set based on [3, 4, 9, 10] |
| $\omega_s$ | Hospitalization rate of asymptomatic infectious | 0.5 (per day) | Set based on [3, 4, 9, 10] |
| $c$ | Percentage of latently infected individuals that is traced and isolated immediately. | 20% | Estimated based on [42][d] |

[a]Estimated based on testing data of arriving truck drivers that was between 1000 and 2000 per day.

[b]Estimated based on the Uganda Bureau of Statistics data that of the 43 million Ugandans, 15 million are school going, >75% of adults engaged in informal sector and the strictness of lockdown measures the initially left only essential workers of approximately 4 million to mingle.

[c]Assumed based on the report that 530 health workers had been infected with 6 fatalities by 30 September 2020.

[d]Estimated based on the reported number of new infections reported among isolated individuals in MoH press briefings.

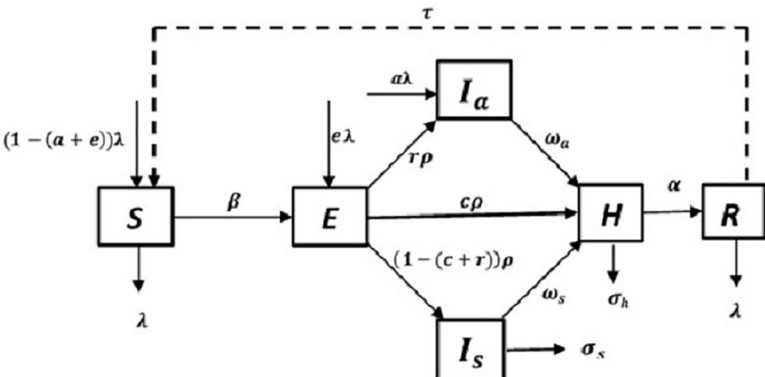

**Fig 1. A compartmental model for fatal COVID-19 transmission dynamics that incorporates arrival of infected and susceptible individuals and departure of susceptible and/or recovered individuals and incorporates contact tracing and isolation of suspected contacts of infectious individuals.** The boxes represent population sub-groups while the arrows indicate the flow of individuals. The dashed arrow presents a yet-to-be resolved aspect on nature, if any, of immunity conferred by the disease upon recovery.

infected per day, $\tau R$ represents the number of previously immune individuals that lose their disease-induced immunity to become susceptible per day and $\lambda S$ represents the number of departing susceptible individuals per day. In Eq (3.2), the term $e\lambda N$ represents the number of arriving latently infected individuals per day while $\rho E$ represents the number of latently infected individuals that progress to become infectious per day. In Eq (3.3), the term $a\lambda N$ represents the number of arriving asymptomatically infectious individuals per day, $r\rho E$ represents the number of latently infected individuals that become asymptomatically infectious per day and $\omega_a I_a$ represents the number of asymptomatically infectious individuals that are hospitalized per day. In Eq (3.4), the term $(1-(c+r))\rho E$ represents the number of latently infected individuals that become symptomatically infectious per day, $\sigma_s I_s$ represents the number of symptomatically infectious individuals that succumb to the disease before being hospitalized per day and $\omega_s I_s$ represents the number of symptomatically infectious individuals that are hospitalized per day. In Eq (3.5), the term $c\rho E$ represents the number of latently infected individuals that were traced and isolated prior to being hospitalized upon testing positive per day, $\omega_a I_a$ represents the number of asymptomatically infectious individuals that are hospitalized per day, $\omega_s I_s$ represents the number of symptomatically infectious individuals that are hospitalized per day, $\sigma_h H$ represents the number of hospitalized individuals that succumb to the disease per day while $\alpha H$ represents the number of hospitalized individuals that recover from the disease per day. Lastly, in Eq (3.6), the term $\lambda R$ represents the number of departing recovered individuals per day and the other two terms are described above.

## 3. Model analysis and simulations

### 3.1 Qualitative analysis

The model was analyzed qualitatively to generate insightful thresholds that can inform public health policy decisions. Under this analysis, the biological feasibility of the model was assessed, the expression for the basic reproduction number ($R_0$) defined as the secondary infections generated by a typical COVID-19 infectious individual when introduced in a naïve population was derived using the next generation matrix method [32] and equilibrium points were determined and their stability analyzed. Summarized key results of these analyses are presented under the results' section and the detailed approaches and all other findings are included in the S1 File.

### 3.2 Quantitative analysis and model simulations

Intervention scenarios were analyzed based on their impact on two selected disease burden indicators namely, the number of hospitalized and undetected infectious cases in the community. Unless stated otherwise, in each scenario; all other parameters were kept at their default values, Day 58 was assumed to be the initiation moment of the assessed additional interventions where applicable and, all simulated scenarios are for the Ugandan population of 43 million with the immediately implemented intervention measures estimated to have initially left 10% of the population susceptible (this percentage representing essential service providers and possible not compliant cases), with this proportion increasing to 30% after 58 days due to easing and complacency. The model was implemented both in Mathematica® 12.0 (Wolfram Research, Inc.) and MatLab software and the simulated period was 365 days.

**3.2.1 Intervention scenarios analysis and possibility of second epidemic wave.** *The impact of lockdown effectiveness and easing.* Here, three scenarios differing in the level of susceptible proportion after 58 days were assessed. Scenario A1 involved maintaining the proportion at 10% while in A2 and A3, the proportion is increased to 30% and 50% respectively. Note

that under this analysis, the other implemented measures such as self-isolation, social distancing and the stay-home-stay-safe campaigns are also implicitly investigated.

*The dynamic impact of constant external disease pressure.* The effect of reducing the number of imported infectious and latently infected cases (after 58 days of reporting first case) by; 50% in Scenario A2, completely eliminating all imported cases in Scenario B2 and Scenario C2 in which no infectious individuals are imported and only 10% latently infected individuals are imported.

*The possibility of a second wave of infection.* Starting with a (hypothetical) situation in which the epidemic has been brought under control with the number of hospitalized individuals reducing significantly during the simulated period, we investigate the possibility of having a second wave upon partial easing on the implemented lockdown measures. This simulation was intended to determine when it would be "safe" to allow up to 75% of the population to resume their normal activities without having to worry about the possibility of having a second wave of infection that can be initiated by the few remnant infectious individuals in the community. Two scenarios in which the proportion susceptible is increased from 30% that was initiated at 58 days to 75% after 210 and 300 days post first case reporting in Uganda are simulated.

**3.2.2 Sensitivity analysis.** Model robustness to and the impact of the various parametric inputs is evaluated using the Latin Hypercube Sampling scheme that was introduced by Marino et al. [33]. This method is based on the Monte Carlo type of sampling methods that caters for unbiased estimates. Since each parameter is sampled independently, a sample size of $N = 300$ (set to this value to guarantee precision of the outcomes) has been divided into $N$ equal probability intervals and the sampling is based on a uniform probability density function, given the novelty of the disease. The model parameters are assessed and ranked based on their impact on the dynamics of the symptomatic infectious individuals since this number is likely to be one of the key determinants of the epidemic burden.

# 4. Results

In this section, we present results from the different analyses of the model.

## 4.1 On qualitative analysis of the model

The basic reproduction number is given by $R_0 = R_1 + R_2 + R_3$, where $R_1 = \frac{b\beta(1-c-r)}{\sigma_s+\omega_s}$ is the reproductive rate of symptomatic individuals, $R_2 = \frac{b\beta rq}{\omega_a}$ is the reproductive rate of asymptomatic individuals and $R_3 = \frac{bg\beta((c+r)\sigma_s+\omega_s)}{(\alpha+\sigma_h)(\sigma_s+\omega_s)}$ is the reproductive rate of hospitalized individuals. Additionally, $\frac{1}{\omega_a}$ is the time that an individual spends in the asymptomatic class, $\frac{1}{\sigma_s+\omega_s}$ is the time that an individual spends in the symptomatic class and $\frac{1}{(\alpha+\sigma_h)}$ is the time a hospitalized individual spends in hospital.

The biological feasibility of the model was guaranteed by proving that all its variables are non-negative at all time t, i.e., the model outputs with positive initial values will remain positive at all time t > 0. On the existence and stability of equilibrium points, the disease free equilibrium point was found to be locally asymptotically stable when $R_0 < 1$ and unstable otherwise. The analysis further revealed that if all infected travelers were denied entry into the country, the disease could only invade at a very low prevalence level if $R_0 > 1$. For further analytical details on these, refer to the S1 File.

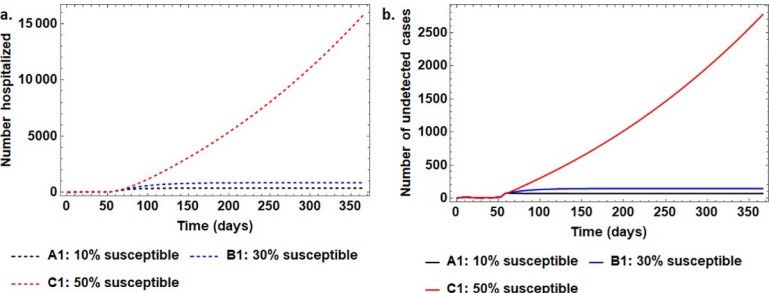

**Fig 2. The effect of reducing the proportion of susceptible population after 58 days since the reporting of the first confirmed COVID-19 case in Uganda.** Scenario A1 involves maintaining the 10% level while the alternative scenarios investigated are B1 involving 30% and C1 involving 50% of the national population of 43 million being susceptible. The different panels depict the time series variation as Panel (a): the number of hospitalized individuals and Panel (b): the number of undetected infectious individuals that are freely living in the communities. In each scenario, all other parameters are kept at their default values.

## 4.2 On the impact of lockdown measures

The results on the assessment impact of reducing susceptible population thereby partially depriving the infection of susceptible individuals are presented in Fig 2. Note that in this analysis, a small percentage of available susceptible population translates to more strict and effective lockdown measures that only avail a few susceptible individuals that may arise from essential workers and the few non-adherent cases.

Overall, the results show that the disease burden is directly and heavily dependent on the proportion of the susceptible individuals available. For example, we observe that, if the available proportion is increased from the initial 10% to 50% after 58 days, within one year, then hospitalized cases will reach very high levels (exceeding 15,000), and the undetected infectious individuals (comprising of the yet-to-be hospitalized asymptomatic and symptomatic infectious individuals) will be high at close to 2500. These results demonstrate that if we ease lockdown by releasing 50% of susceptible population for the Ugandan situation with 3200 hospital beds (of which only 55 were functional ICU as of 2018 [34]) within 150 days, the COVID-19 related hospitalization demand would have already overwhelmed the current resources. Besides over-stretching the health care system, the number of undetected cases that would be living freely in the community, of whom a half would be asymptomatic and hence harder to detect, would further strain disease response efforts of contact tracing. Note that, even in the relatively ideal situations of maintaining only 10% or 30% of susceptible population mingling, the maximum hospitalization requirements would reach 369 and 844 beds respectively within one year, way beyond the number of functional ICU beds.

## 4.3 On the dynamic impact of constant external disease pressure

The findings from the assessment of the imported cases under partially lifted lockdown are presented in Fig 3. Generally, the results reveal that the disease will not be wiped out of Uganda for as long as we still have imported cases no matter how few. We observe in Scenario A2 that reducing the imported cases by half will not be immediately impactful as the disease will continue to grow exponentially for months. Moreover, even in the near perfect scenario (C2) where all asymptomatically infected are blocked along with blocking 90% of arriving latently infected entrants, the disease will persist in the community past the one year simulated period. Even more worrying is the fact that the prevalence level would be higher at higher levels of susceptible population.

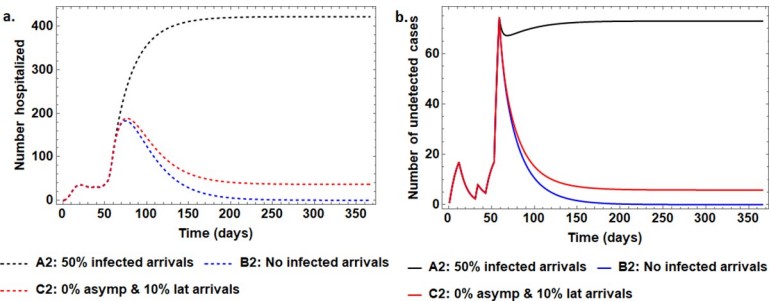

**Fig 3. The effect of reducing the number of imported COVID-19 cases and latently infected individuals by 50% in Scenario A2, are completely eliminated in Scenario B2 and in Scenario C2 there are no asymptomatic infectious and there are only 10% latently infected individuals that are granted entry after 58 days post first case reporting in Uganda.** These measures are implemented together with a partially lifted lockdown where 30% (instead of 10% that is assumed to be in place before the 58 days) of the population is susceptible. The different panels depict the time series variation for, Panel (a): the number of hospitalized individuals and Panel (b): the number of undetected infectious individuals that are freely living in the communities. In each scenario, all other parameter values at maintained at their default values.

## 4.4 On the possibility of a second wave of infection

The question addressed here pertains to determining when and how lockdown should be lifted without posing the risk of having a second wave of infection. Results on this assessment are presented in Fig 4. The model predicted outcomes demonstrate a possibility of observing a second wave of infection upon lifting the lockdown too soon. The simulations reveal that with the current levels of contact tracing, having released up to 75% of susceptible after 210 days would guarantee emergence of a second epidemic wave. It is even more worrying, that enhancing contact tracing would only affect the magnitude and timing of the second wave but cannot prevent it altogether. Thus it can concluded that, even in a fairly ideal situation in which the control efforts in place (e.g., total elimination of imported cases manage to initially curb the spread of the virus, once the lockdown is hurriedly lifted, the yet-to-be detected cases in the community, no matter how few, have potential to start a second wave of infection.

## 4.5 On sensitivity analysis

The findings from the sensitivity analysis are presented in Fig 5. These results could ideally partly inform the process of prioritizing intervention measures as they rank the model

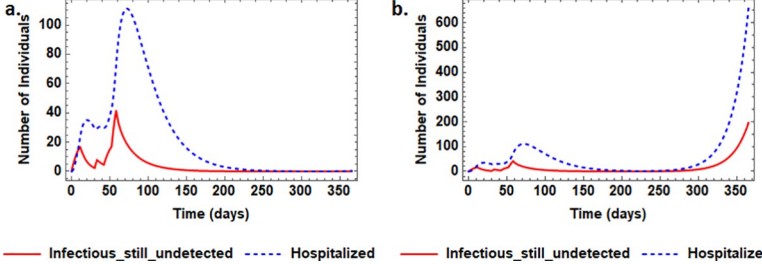

**Fig 4. The possibility of having a second wave of COVID-19 infections upon easing lockdown measures with no hospital acquired infections and after 58 days, there is assumed to be an initial 30% ease in the lockdowns and no more imported cases.** Panel (a) shows easing of lockdown from 10% to 30% susceptible after 58 days and further easing to 75% after 300 days of lockdown and could be sooner if contact tracing efforts were intensified. Panel (b) is a case of easing lockdown from 10% to 30% after 58 days followed by 75% susceptible after 210 days of lockdown–a case of too many and too soon. In each scenario, all other parameter values are maintained at their default values.

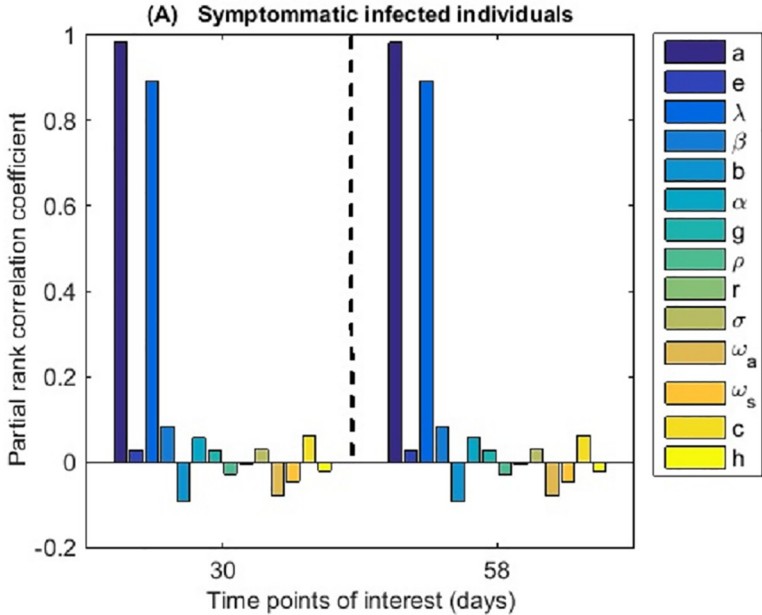

**Fig 5. Pearson's Rank Correlation Coefficients (PRCCs) tornado simulation of the model's most sensitive parameters sampled 300 times using the Latin hypercube sampling scheme.** Sensitivity variation with respect to time is analyzed at two time points, 30 days and 58 days that could be used as initialization points or start times for the intervention measures. The order of appearance of parameters in the figure legend follow the left to right order in labeling the corresponding bars. For example, the topmost label corresponds to the first bar in the figure and so on.

parameters based on their impact on one of the key indicators of the epidemic burden. From the sensitivity analysis, the most impactful parameters identified include the arriving proportion of infected individuals ($a$), the per capita recruitment rate ($\lambda$), the disease transmission rate ($\beta$), the proportion of susceptible population available for infection ($b$), and the efforts of contact tracing and subsequent isolation of latently infected individuals ($c$) among others.

## 5. Discussion

Findings of this study show that the immediately implemented measures by the Government of Uganda averted thousands of cases that would have overstretched the health system within a couple of months (Fig 2). This would have consequently affected the quality of treatment offered leading to more fatalities and hospital acquired infections due to fatigue and laxity. The results in Fig 2 further show that even with only a 10% susceptible fraction, the number of hospitalizations would reach as high as 369 individuals within a year. If 5–7% of these require intensive care admission as estimated for Wuhan [35], the capacity of the available functional ICU bed facilities in Uganda which stood at only 55 as of 2018 [34] would be exceeded.

On imported cases, results show that without significantly altering the current situation, measures on partial lockdowns and use of masks are insufficient to stop COVID-19 and as such the disease will remain in the population (Fig 3). In all the assessed scenarios the disease would be wiped out in the case where there are no infected arrivals beyond the first 58 days and in this case the disease would be wiped out within 270 days as seen from Fig 3. This impact of imported cases is also emphasized by the findings from the sensitivity analysis. Currently, these cases are being regulated through enforcement of mandatory screening for clinical signs and testing at border points. This screening faces a challenge of reagent limitation, imperfect test accuracy, arrival of asymptomatic and latently infected individuals that may pass as false

negatives during screening as well as the porosity of some of the national borders. Thus, adoption of alternative less-risky means of essential cargo delivery (e.g., by rail and ship services) combined with quarantining of all entrants for a duration not shorter than the incubation period should be enforced. Besides improving test accuracy and sample handling, the number of false negatives can also be minimized by ensuring that individuals do not get infected so close to being tested as the disease is not detectable until the latent period is surpassed. Thus, avenues of infection at testing points need to be eliminated. With the worrying situation of increased reported cases in our neighboring countries, the impact of Uganda's interventions would be greatly affected.

Amidst challenges of social-economic impact of COVID-19, agitation of lifting lockdown may downplay the impact of intervention measures and the study findings highlight the importance of optimal timing and magnitude of lockdown easing (Fig 4). Effective phased-out ease of lockdown needs to be well studied and executed. The results in Fig 4 reveal that, even in a fairly ideal situation with no new arrivals of imported cases, once the lockdown is hurriedly lifted to a 75% level, the yet-to-be detected cases in the community have potential to start a second and more disastrous epidemic wave. Note however that with enhanced surveillance and contact tracing, gradual easing by releasing smaller percentages of susceptible individuals from lockdown can still be safely executed sooner than the optimum 300 days for the 75%. On the importance of a phased-out release of lockdown restrictions, a modeling study parametrized for the United Kingdom reported that the optimum strategy was to release approximately half of the population 2 to 4 weeks from the end of an initial infection peak and then wait another 3 to 4 months to release everyone [22].

Sensitivity analysis results (Fig 5) reveal that recruitment of infected truck drivers, lockdown measures together with enhanced contact tracing may have averted many subsequent hospitalizations thereby minimizing disease burden on the already strained health-care system. Contact tracing can further be improved through increasing manpower, improving efficiency follow-up tools e.g., registry, public awareness and incentives to ensure voluntary/self-reporting and empowering of local response teams. Although the parameter influencing the number of hospital acquired infections was of relatively low impact on the number of symptomatic cases in the sensitivity analysis, field experience with handling COVID-19 in other countries (note that Uganda did not report any hospital acquired infection for the first two months of her epidemic) as well as diseases such as Ebola in Uganda dictates that this should be maintained at a bare minimum to ensure that health workers do not shun their duties. It is clear that hospital acquired infections go beyond merely increasing the number of cases, their mitigation should be given high priority. This can be achieved through ensuring appropriate use of properly fitting personal protective equipment, providing refresher training to the frontline medical personnel and ensuring adequate staffing at the treatment and sample collection centers among others.

There are some limitations to our analysis that may either arise from the assumptions of the model or its parametrization since we used biologically plausible parameters based on current evidence yet some may be refined as more comprehensive data become available. For example, the COVID-19 induced mortality rate derived for this study based on data from Uganda is relatively lower than global values (e.g., those reported in [36]) and this may be validated in future. The other parameter that has been reported to vary widely across countries is the percentage of asymptomatic individuals. Several studies have estimated this proportion using data from different sources including hospitals and cruise ships and their estimated values have ranged from 18% to 81% [37–41].

Given the demonstrated coupling of Uganda's disease dynamics with those in her neighboring countries, network based modeling approaches would be better suited to assess potential

impact of the coupling pathways on the disease dynamics. Although we assumed constant external disease pressure in this study, we recommend that future studies adopt the network modeling approach whenever relevant data to substantiate those network based models is available. On lockdown modeling, we assumed that a percentage of the susceptible population was isolated by the lockdown restrictions and hence impossible to become infected. This assumption is justified by the strictness in enforcing these measures in Uganda. However, for relatively lax implementation of the measures, alternative approaches such as that involving splitting the population into the quarantine and non-quarantine groups as done by [22] could be more suitable.

In all, results highlight importance of strict adherence to SOPs, minimization of complacency as well as maintenance of partial lockdowns until a point when the regional and even global situation improves. Given the scantiness of details on the pathogen characteristics (e.g., the duration of the disease-induced immunity and consequently herd-immunity and, infectiousness of asymptomatic individuals) and the epidemiological information, return to full activity may have to wait until when an effective vaccine and/or drug is found and used massively.

## Supporting information

**S1 File.**
(PDF)

## Acknowledgments

The Authors acknowledge the Ministry of Health for the data on COVID-19.

## Author Contributions

**Conceptualization:** Joseph Y. T. Mugisha, Joseph Ssebuliba, Juliet N. Nakakawa, Cliff R. Kikawa, Amos Ssematimba.

**Data curation:** Joseph Ssebuliba, Juliet N. Nakakawa, Cliff R. Kikawa, Amos Ssematimba.

**Formal analysis:** Joseph Y. T. Mugisha, Joseph Ssebuliba, Juliet N. Nakakawa, Amos Ssematimba.

**Funding acquisition:** Joseph Y. T. Mugisha.

**Investigation:** Joseph Y. T. Mugisha, Joseph Ssebuliba, Juliet N. Nakakawa, Amos Ssematimba.

**Methodology:** Joseph Y. T. Mugisha, Joseph Ssebuliba, Juliet N. Nakakawa, Cliff R. Kikawa, Amos Ssematimba.

**Project administration:** Joseph Y. T. Mugisha.

**Software:** Joseph Ssebuliba, Juliet N. Nakakawa, Amos Ssematimba.

**Validation:** Amos Ssematimba.

**Writing – original draft:** Amos Ssematimba.

**Writing – review & editing:** Joseph Y. T. Mugisha, Joseph Ssebuliba, Juliet N. Nakakawa, Cliff R. Kikawa.

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
