## [Decision Letter · Decision Letter 0]

8 Oct 2020

PONE-D-20-24477

Mathematical modelling of COVID-19 transmission dynamics in Uganda: implications of complacency and early easing of lockdown

PLOS ONE

Dear Dr. Ssematimba,

Thank you for submitting your manuscript to PLOS ONE. After careful consideration, we feel that it has merit but does not fully meet PLOS ONE’s publication criteria as it currently stands. Therefore, we invite you to submit a revised version of the manuscript that addresses the points raised during the review process.

In particular, please pay attention to the comments by Reviewer #2, on the origin and cause of the 'second wave'. Is this an endogenous event (generated by the model without changing any of its parameters), or is this an exogenous event (caused by factors beyond the model, and typically treated as changes in its parameters)?

We look forward to receiving your revised manuscript.

Kind regards,

Siew Ann Cheong, Ph.D.

Academic Editor

PLOS ONE

Journal Requirements:

Reviewers' comments:

Reviewer's Responses to Questions

**Comments to the Author**

1. Is the manuscript technically sound, and do the data support the conclusions?

Reviewer #1: Yes

Reviewer #2: Partly

2. Has the statistical analysis been performed appropriately and rigorously? 

Reviewer #1: Yes

Reviewer #2: No

3. Have the authors made all data underlying the findings in their manuscript fully available?

Reviewer #1: Yes

Reviewer #2: Yes

4. Is the manuscript presented in an intelligible fashion and written in standard English?

Reviewer #1: Yes

Reviewer #2: Yes

5. Review Comments to the Author

Reviewer #1: Abstract should be structural manner including of subsection like background, methods, results, discussion.

The introduction section produces redundant information on COVID-19 background which could be no interest for readers. Rather it is recommended to present related modelling works. Some of works the suggestive

a) COVID-19 disease outbreak forecasting of registered and recovered cases after sixty-day lockdown in Italy: A data driven model approach

b) A Simulation of a COVID-19 Epidemic Based on a Deterministic SEIR Model

c) Modified SEIR and AI prediction of the epidemics trend of COVID-19 in China under public health interventions

Clarity of figures is missing. Please enhance the figure aspect ratios

Reviewer #2: 1. A standard presentation of this kind of model usually includes derivation of R¬0, the basic reproduction number, equilibrium points, and stability analysis and these should be done mathematically (not through simulation or numerically). Since the focus of this paper is the dynamics that are mainly observed through simulations, the authors can present the mathematical aspect in a very brief way. But this is very important for this kind of modeling.

2. Possibility and impact of a second wave is not clear. I do not understand how this second wave is accommodated in the model. If the authors intend to provide a time line when this second wave might hit, that would be great. However, the objective and presentation can be made much more explicit. This possibility of the second wave is an important issue in this paper. So clear understanding and prediction of time and extent, if possible, should be presented and justified properly.

3. There is no justification of the sample size N=300 in sensitivity analysis.

4. Few minor English corrections are required, like at page 95: Write “since” instead of “from”, etc.

6. PLOS authors have the option to publish the peer review history of their article (what does this mean?). If published, this will include your full peer review and any attached files.

Reviewer #1: No

Reviewer #2: **Yes: **Indranil Mukhopadhyay

---

## [Author Response · Author response to Decision Letter 0]

22 Oct 2020

Comments to the Author

Reviewer #1: 

Abstract should be structural manner including of subsection like background, methods, results, discussion.

This observation is appreciated although only slight amendments have been made. Generally, the abstract was drafted in such a way that the different paragraphs capture content for the different subsections albeit without specific headings. There is one paragraph each for Background, Materials and Methods, Results and Discussion in accordance with PLOS ONE journal guidelines to authors.

The introduction section produces redundant information on COVID-19 background which could be no interest for readers. Rather it is recommended to present related modelling works. Some of works the suggestive

a) COVID-19 disease outbreak forecasting of registered and recovered cases after sixty-day lockdown in Italy: A data driven model approach

b) A Simulation of a COVID-19 Epidemic Based on a Deterministic SEIR Model

c) Modified SEIR and AI prediction of the epidemics trend of COVID-19 in China under public health interventions

The introduction section has been improved upon and paragraphs dedicated to COVID-19 modelling have been introduced as suggested by the reviewer and the proposed articles have been utilized. These edits are captured on page 4 lines 68-77 as “…The role of compartmental mathematical models in predicting COVID-19 dynamics under varying interventions has been demonstrated. Examples of such studies include, Yang et al. [19] who effectively predicted the epidemic peaks and sizes in China using an approach that integrated population migration and COVID-19 epidemiological data into a Susceptible-Exposed-Infectious-Removed (SEIR) model together with an artificial intelligence approach. Also, Carcione et al. [21] demonstrated how mathematical models can be used to generate meaningful insights amidst epidemiological and etiological data challenges. They simulated a COVID-19 epidemic for Italy based on an SEIR model and they emphasized the importance of assessing the effectiveness of lockdowns on epidemic burden. …”

Clarity of figures is missing. Please enhance the figure aspect ratios

The quality of the figures has been improved upon.

Reviewer #2: 

1. A standard presentation of this kind of model usually includes derivation of R¬0, the basic reproduction number, equilibrium points, and stability analysis and these should be done mathematically (not through simulation or numerically). Since the focus of this paper is the dynamics that are mainly observed through simulations, the authors can present the mathematical aspect in a very brief way. But this is very important for this kind of modeling.

The qualitative analysis of the model was done earlier but the write of the paper was hinged more on the quantitative analysis give the original objective. However, on the reviewer’s recommendation, we have now reintroduced the content on the qualitative analysis both in the main text and as supplementary information. The detailed analysis steps and all other results have been included as supporting information in file S1 for completeness.

We now have a brief description of methods in Section 3.1 page 9 lines 167-174 as “…Qualitative analysis of the model: The model was analyzed qualitatively to generate insightful thresholds that can inform public health policy decisions. Under this analysis, the biological feasibility of the model was assessed, the expression for the basic reproduction number (R0) defined as the secondary infections generated by a typical COVID-19 infectious individual when introduced in a naïve population was derived using the next generation matrix method [27] and equilibrium points were determined and their stability analyzed. Summarized key results of these analyses are presented under the results’ section and the detailed approaches and all other findings are included in the supporting information file S1….” 

A summary of key results on this analysis has been introduced in section 4.1 page 11 in lines 224-236 as “…On qualitative analysis of the model: The basic reproduction number is given by R_0=R_1+R_2+R_3, where R_1=(b β(1-c-r))/(σ_s+ω_s ) is the reproductive rate of symptomatic individuals, R_2= (b β r q)/ω_a is the reproductive rate of asymptomatic individuals and R_3= (b g β ((c+r) σ_s+ω_s))/((α+σ_(h ) )(σ_s+ω_s)) is the reproductive rate of hospitalized individuals. Additionally, 1/(ω_a ) is the time that an individual spends in the asymptomatic class, 1/(σ_s+ω_s ) is the time that an individual spends in the symptomatic class and 1/((α+σ_(h ) ) ) is the time a hospitalized individual spends in hospital. 

The biological feasibility of model system was guaranteed by proving that all its variables are non-negative at all time t, i.e., the model's outputs with positive initial values will always remain positive at all time t > 0. On the existence and stability of equilibrium points, the disease free equilibrium point was found to be locally asymptotically stable when R0 < 1 and unstable otherwise. The analysis further revealed that if all infected travelers were denied entry into the country, then the disease could only invade at a very low endemic level if R0 > 1. For further analytical details on these, refer to the supporting information file S1….” 

2. Possibility and impact of a second wave is not clear. I do not understand how this second wave is accommodated in the model. If the authors intend to provide a time line when this second wave might hit, that would be great. However, the objective and presentation can be made much more explicit. This possibility of the second wave is an important issue in this paper. So clear understanding and prediction of time and extent, if possible, should be presented and justified properly.

There was lack of clarity in presenting aspects about the possibility and impact of a second wave and we have now improved upon the flow of content on this aspect and the manuscript in general. 

On the origin and cause of the 'second wave', note that this study in part aimed to evaluate the impact of the intervention measures that were implemented by the Government of Uganda that included lockdown of almost all nonessential activities in the whole country. This study aimed to generate insight into how best the instituted lockdown could be lifted (i.e., when and what percentage to be optimally lifted from lockdown) while averting the second wave. Technically, the predicted second wave is an exogenous event resulting specifically from a hurried partial easing of lockdown when there still are untraced infectious individuals freely living in the community no matter how few. Embedded in the parameter b (i.e. percentage of susceptible individuals that is available) that captures the proportion of people that is out of lockdown and is therefore “mingling” or interacting freely.

We have added the following in the methods section on page 10 lines 199-202 as “…This simulation was intended to determine when it would be “safe” to allow up to 75% of the population to resume their normal activities without having to worry about the possibility of having a second wave of infection that can be initiated by the few remnant infectious individuals in the community…” 

In the Discussion section we mention the that the second wave was a consequence of easing lockdown on page 17 lines339-342 as “The results in Figure 5 reveal that, even in a fairly ideal situation with no new arrivals of imported cases nor the occurrence of hospital acquired infections, once the lockdown is hurriedly lifted to a 75% level, the yet-to-be detected cases in the community have potential to start a second and more disastrous epidemic wave.…”

3. There is no justification of the sample size N=300 in sensitivity analysis.

Note that N denotes the sample size and the key requirement it that its value be greater than the number of parameters varied plus 1 (i.e. j+1) but recommended to be set to a large enough value to guarantee precision while bearing in mind the need for maintaining computer run time and efficiency. N=300 gave us both precision and efficiency. We have added a justification in the main text page 11 lines 218-219 as “…a sample size of N=300 (set to this value to guarantee precision of the outcome)…”

4. Few minor English corrections are required, like at page 95: Write “since” instead of “from”, etc.

This and other textual edits have been made.

---

## [Decision Letter · Decision Letter 1]

23 Nov 2020

PONE-D-20-24477R1

Mathematical modeling of COVID-19 transmission dynamics in Uganda: Implications of Complacency and Early Easing of Lockdown

PLOS ONE

Dear Dr. Ssematimba,

Thank you for submitting your manuscript to PLOS ONE. After careful consideration, we feel that it has merit but does not fully meet PLOS ONE’s publication criteria as it currently stands. Therefore, we invite you to submit a revised version of the manuscript that addresses the points raised during the review process.

While it is clear that the manuscript has been improved from the original submission, Reviewer #1 is concerned that the authors did not address specific comments from the first review. We understand that PLOS ONE gives a rather short time to make revisions, but this is just a standard procedure to shorten the time to acceptance and publication. Should you need more time for major revisions, please do not hesitate to make the request. More importantly, because of the COVID-19 situation, Reviewer #2 could no longer assist in the review of the revised manuscript. We are very fortunate to have Reviewer #3 agreeing to help, and to produce a very detailed report on the revised manuscript. Unfortunately, Reviewer #3 has identified further major weaknesses. We advise the authors to take their time to properly address these, before submitting another revision.

We look forward to receiving your revised manuscript.

Kind regards,

Siew Ann Cheong, Ph.D.

Academic Editor

PLOS ONE

Reviewers' comments:

Reviewer's Responses to Questions

**Comments to the Author**

1. If the authors have adequately addressed your comments raised in a previous round of review and you feel that this manuscript is now acceptable for publication, you may indicate that here to bypass the “Comments to the Author” section, enter your conflict of interest statement in the “Confidential to Editor” section, and submit your "Accept" recommendation.

Reviewer #1: (No Response)

Reviewer #3: (No Response)

2. Is the manuscript technically sound, and do the data support the conclusions?

Reviewer #1: Yes

Reviewer #3: No

3. Has the statistical analysis been performed appropriately and rigorously? 

Reviewer #1: Yes

Reviewer #3: N/A

4. Have the authors made all data underlying the findings in their manuscript fully available?

Reviewer #1: Yes

Reviewer #3: Yes

5. Is the manuscript presented in an intelligible fashion and written in standard English?

Reviewer #1: Yes

Reviewer #3: Yes

6. Review Comments to the Author

Reviewer #1: I wondering that authors did not address the specific comments rasied by reviewer that might considered as not a good practice.

Abstract section should has to mention the sub section and we can not assume that readers will automatically understand the authors view.

Authors has to address the established data modelling approches in non ease of lockdown and compare it with some spcial case studies like Italy and other european countries (COVID-19 disease outbreak forecasting of registered and recovered cases after sixty-day lockdown in Italy: A data driven model approach).

Some areas spell check and grammetical errors has to address

Best wishes

Reviewer #3: The manuscript utilises a compartmental SEIHR framework to assess the disease dynamics in Uganda, and consider how to tackle future infection possibilities. Current manuscript revisions have already improved the manuscript well. While I feel the work has great potential to advance the understanding of disease dynamics within Uganda, I have substantial concerns with the current model formulation, and the conclusions the authors draw.

The primary issue is that the authors attempt to, arguably, do too much with the model, and in doing so each conclusion suffers. I list below my general concerns with the model formulation, and which scenarios are ill-posed for the current model. In general I would urge the authors to restrict their scope, and tweak the model to draw stronger conclusions on a specific research question or two. I feel the model may be best suited with minor adjustment to assessing “The possibility of a second wave infection”, or “The impact of lockdown measures”, while it is currently not an appropriate model for assessing “The impact of magnitude of imported cases” or “Impact of hospital acquired infections”.

The Introduction is very strong and suitably frames the work.

General Concerns

My initial concerns are with the model parameters (Table 1). Many parameters cite the Ugandan data portal (reference [4]) for their values, however I cannot locate these model parameters, nor can I infer them from Ugandan case data. Some parameters are also very far from similar parameter estimates in the literature. For example, a disease-induced mortality rate of 0.001 is far low than estimates elsewhere in the literature. The hospital mortality rate of 0.0001 is even more peculiar. See for example, Baud et al. (2020) “Real estimates of mortality following COVID-19 infection.”.

An estimate of 40% asymptotic individuals could hold true, there are multiple studies citing closer to 20% (e.g. Mizumoto et al.), however some more recent studies predict as high as 80% (Day M. “Covid-19: four fifths of cases are asymptomatic, China figures indicate.”). Mentions of such literature should be included.

Similarly recovery rate, traced-and-isolated rates ( c ), infectivity in hospitals, need closer scrutiny. All model parameters need more source citations.

Parameter q can be removed if it is not used in the model.

Moving on to the model itself, a major concern is that individuals who are contacted via test-and-trace are moved to the hospitalisation class (If I understood correctly). This is a major oversight as it will be a significant move of individuals, having a considerable impact on the resulting dynamics, and invalidating any assessment of total hospitalisation numbers (such as figures 2,3,5). This estimation of hospitalised individuals is a major use of these models, and so the assessment must be rigorous.

I am also sceptical of the implementation of imported cases. Having this influx of individuals (and I don’t fully understand how these imported cases subsequently leave the population pool in an equal amount) be dependent on the host population size (N) doesn’t make sense. In general, one could perhaps argue for a constant population (in/out) from the infected class, but this formulation is not suitable for then drawing conclusions on the dynamic impact of imported cases from neighbouring countries. It would be better to develop a model of multiple interacting SEIHR networks for this. I would recommend either removing this aspect or better framing this as “the dynamic impact of constant external disease pressure”.

I would also be interested in more information on the exact lockdown rules in Uganda to defend the choice of lockdown modelling. The authors have a percentage of the population which is made impossible to become infected due to lockdown. This is appropriate for strict lockdown systems, such as that in Cyprus, where individuals may not leave home without permission, but for systems such as the UK where individuals may leave home freely for essential trips, a far reduced transmission rate may be more appropriate than a flat impossibility of transmission (e.g. Rawson et al. “How and when to end the COVID-19 lockdown”).

When the model is introduced (from L102), it is very difficult to interpret the model initially, needing to jump between table 1, figure 1, equations (1)-(5), and brief paragraphs. I would instead urge presenting each equation, one-by-one, and explaining specifically each term (and parameter) in each equation. This may seem excessive, but makes interpreting a model considerably easier.

The above issues then raise concern with the interpretation of results:

4.2 – This section seems the most powerful to focus on, and could be expanded by comparing and fitting to Ugandan case data. The current primary result is not particularly useful or enlightening, less people in lockdown = more people infected. More interesting would be to then expand this to ask “how strict does a lockdown need to be to not exceed hospital capacity?” “When should a lockdown be ended?” etc. The current metric of hospitalisations is incorrect due to the above test-and-trace classification issues.

4.3 – I currently don’t think this model formulation is powerful enough to answer questions on the impact of imported cases. The result that “COVID remains endemic” is due precisely to the model formulation. The constant influx of infection from these cases ensures an endemic outbreak. If the authors wanted to investigate the impact of imported cases then this influx should be variable. Then the model can answer questions such as “When should imports from country A be allowed based on their case rates?”.

4.4 – This section will be insightful, however the above concerns on model formulation and parameterisation need to be addressed first.

4.5 – I don’t think the model is suitable for assessing the impact of hospital-acquired infections currently. This is an interesting dynamic, but would benefit from a more bespoke model, with separate healthcare compartmentalisations. The current hopsitalisation framework (see above) invalidates the current results.

4.6 – I’m not sure the text here actually correlates with the sensitivity analysis of Figure 6. The text states that a, e, b, and g are most influential. However Figure 6 seems to infer that a and lambda are the most impactful, and the others have minor impact. Certainly g and e seem of low impact. Also, the bins of Figure 6 should be marked with their respective parameters. The current colour scheme is hard to distinguish.

Minor Comments:

L51/L52 – “measures reduced the susceptible population to about 10%”, this is important, is there a citation for this?

L129 – Does this mean that asymptomatic individuals are hospitalised? I assume this is due to the test-and-trace, this needs to be changed.

L266 – What is meant by undetected infectious here? Mentioned specifically which model variable you mean when referenced (e.g. I/E)

The model is good and I think it is of critical importance to investigate country-specific disease trajectories, the scope and details of the model just need to be refined. I look forward to seeing a more refined revision and believe it will make a strong contribution when finished.

7. PLOS authors have the option to publish the peer review history of their article (what does this mean?). If published, this will include your full peer review and any attached files.

Reviewer #1: No

Reviewer #3: **Yes: **Thomas Rawson

---

## [Author Response · Author response to Decision Letter 1]

15 Dec 2020

15 December 2020

Revision of manuscript number PONE-D-20-24477R1 titled “Mathematical modeling of COVID-19 transmission dynamics in Uganda: Implications of Complacency and Early Easing of Lockdown” 

On behalf of the co-authors, we appreciate all your efforts devoted towards improving this manuscript. Thank you so much!

Review Comments to the Author

Reviewer #1: I wondering that authors did not address the specific comments rasied by reviewer that might considered as not a good practice.

Abstract section should has to mention the sub section and we can not assume that readers will automatically understand the authors view.

As requested, we have introduced section headers in the abstract on page 1 in lines 14-21.

Authors has to address the established data modelling approches in non ease of lockdown and compare it with some spcial case studies like Italy and other european countries (COVID-19 disease outbreak forecasting of registered and recovered cases after sixty-day lockdown in Italy: A data driven model approach).

We have addressed this by introducing content comparing pre- and post- intervention measures in the introduction as on page 4 in lines 80-84 as “… Relatedly, Chintalapudi et al.[21] concluded basing on outcomes from their data driven model analysis that if the Italian government had extended the March 2020 country lockdown and self-isolation measures by 60 days, the number of registered cases would decrease by 35% while those recovered would increase by 66%. All these approaches demonstrate the utility of mathematical models in assessing the impact of intervention measures on COVID-19 dynamics. …” Relatedly, our study assesses the impact of lockdown through assessing the disease burden (number of new cases and number hospitalized) under different lockdown percentages (10%, 30% and 50%) as depict in Figure 2. 

Following the reviewer advice, we also deemed it prudent to add more related literature. We added content from another relevant study by Rawson et al. (2020) on page 4 in lines 84-87 as “… On using modeling to assess how efficiently lockdowns can be eased in the United Kingdom, Rawson et al. [22] applied an optimal control framework to an adapted SEIR model framework and found that the optimal strategy was a phased-out release strategy. …” 

We also added related content in the discussion section on page 17 in lines 352-355 as “… On the importance of a phased-out release of lockdown, a modeling study parametrized for the United Kingdom reported that the optimum strategy was to release approximately half of the population 2 to 4 weeks from the end of an initial infection peak and then wait another 3 to 4 months to release everyone [Rawson et al. 2020]. …”

Some areas spell check and grammetical errors has to address

These have been addressed.

Best wishes

Reviewer #3: The manuscript utilises a compartmental SEIHR framework to assess the disease dynamics in Uganda, and consider how to tackle future infection possibilities. Current manuscript revisions have already improved the manuscript well. While I feel the work has great potential to advance the understanding of disease dynamics within Uganda, I have substantial concerns with the current model formulation, and the conclusions the authors draw.

The primary issue is that the authors attempt to, arguably, do too much with the model, and in doing so each conclusion suffers. I list below my general concerns with the model formulation, and which scenarios are ill-posed for the current model. In general I would urge the authors to restrict their scope, and tweak the model to draw stronger conclusions on a specific research question or two. I feel the model may be best suited with minor adjustment to assessing “The possibility of a second wave infection”, or “The impact of lockdown measures”, while it is currently not an appropriate model for assessing “The impact of magnitude of imported cases” or “Impact of hospital acquired infections”.

The Introduction is very strong and suitably frames the work.

We agree that trimming and refocusing the content tremendously improves that quality of the manuscript and as suggested by the reviewer, the manuscript has been refocused to address three specific questions namely; 1) assessing the potential impact of the implemented lockdown measures, 2) the dynamic impact of constant external disease pressure (phrasing suggested by the reviewer in the later comments), and 3) the possibility of the second wave of infection. We also maintain the section on sensitivity analysis.

General Concerns

My initial concerns are with the model parameters (Table 1). Many parameters cite the Ugandan data portal (reference [4]) for their values, however I cannot locate these model parameters, nor can I infer them from Ugandan case data. 

For reliability of the study outcomes in informing Uganda policy interventions to manage the disease, it was necessary that the study uses localized parameters whenever possible. However, as was the case with other earlier modeling studies, only scanty information about the pathogen and the disease in general was available and more is just being generated as the pandemic progresses. Therefore, some of the earlier simulation studies utilized the limited data together with sensitivity analyses to infer disease dynamics under varying intervention scenarios. It is hoped that as more knowledge about the pathogen and its transmission pathways in generated, the models will be updated accordingly.

For our Ugandan case, not all required parameters have been documented in citable literature and this was a hurdle to the current study. For example, besides the number of new cases, recoveries and deaths that are mandatorily documented and forwarded to WHO among other stakeholders, the other parameters that cite the Ministry of Health data (reference 4) are mainly inferred or derived from the routine addresses by the personnel from the Ministry of Health, Uganda virus research institute and the office of the presidency among others and are based on observational data. Examples of such include the arrival rate of truck drivers (both infected and non-infected), number of returning residents through airports, time spent in hospital for the admitted cases which partly informs the model recovery rate, the proportion of asymptomatic infected individuals etc. In our parameter estimation, content from these routine national addresses and press releases which is often summarized in news outlets was used to infer some of the model parameters. One of such briefings can be found online at https://www.health.go.ug/document/update-on-covid-19-response-in-uganda/ and https://www.health.go.ug/covid/category/press-release/ Studies are being conducted and it is hoped that such new studies will have data from which other more precise parameters can be obtained. For example the recently published Kirenga et al. (2020) “Characteristics and outcomes of admitted patients infected with SARSCoV-2 in Uganda” https://www.ncbi.nlm.nih.gov/pmc/articles/PMC7477797/pdf/bmjresp-2020-000646.pdf validates some of the parameter estimates.

We have checked the literature again and identified some articles that we could cite for some of the parameters in Table 1. For recruitment rate [Ministry of Health -Uganda. UPDATE ON THE COVID-19 OUTBREAK IN UGANDA available online at https://www.health.go.ug/covid/category/press-release/ accessed on 1 December 2020. 2020.], for recovery rate of hospitalized individuals [WHO-China. Report of the WHO-China Joint Mission on Coronavirus Disease 2019, available online at https://www.who.int/docs/default-source/coronaviruse/who-china-joint-mission-on-covid-19-final-report.pdf, accessed 3 June 20202020.] and for percentage of latently infected individuals in community that becomes asymptomatic [Daniel P. Oran A, Eric J. Topol. Prevalence of Asymptomatic SARS-CoV-2 Infection. Annals of Internal Medicine. 2020;173(5):362-7. doi: 10.7326/m20-3012 %m 32491919.]

Some parameters are also very far from similar parameter estimates in the literature. For example, a disease-induced mortality rate of 0.001 is far low than estimates elsewhere in the literature. The hospital mortality rate of 0.0001 is even more peculiar. See for example, Baud et al. (2020) “Real estimates of mortality following COVID-19 infection.”.

The low disease-induced death rate that used in our study is for the Ugandan situation where Uganda did not register any COVID-19 related death in the first four months from the first case (i.e. March 22nd to July 22nd 2020). The first death was registered on 23rd July 2020 and at that point, the cumulative number of cases was 1079 with 971 recoveries. In other countries, using data as of March 1st 2020 (i.e. approx. 4 months (December to March) into the outbreak), Baud et al. (2020) reported mortality rate of 3·6% (95% CI 3·5–3·7) in China and 1·5% [1·2–1·7] outside of China and when adjusted for incubation period delay, mortality rates would be 5·6% (95% CI 5·4–5·8) for China and 15·2% (12·5–17·9) outside of China. Moreover, even as of December 1st 2020, globally there are approximately 63.7m cases, 1.5m deaths and 44.1m recoveries while in Uganda, there have been 20,459 cases, with 205 fatalities and 8989 recoveries. These numbers may crudely indicate both the case-fatality ratio and Recovered-fatality ratio (obtained from considering only the resolved cases) are relatively lower in Uganda. 

For completeness and clarity, we have added the following “justification” also citing Baud et al (2020) in the discussion on page 18 in lines 367-372 as “… There are some limitations to our analysis that may either arise from the assumptions of the model or its parametrization since we used biologically plausible parameters based on current evidence yet some may be refined as more comprehensive data become available. For example, the COVID-19 induced mortality rate derived for this study based on data from Uganda is relatively lower than global values (e.g., those reported in [33]) and this may be validated in future. …”

An estimate of 40% asymptotic individuals could hold true, there are multiple studies citing closer to 20% (e.g. Mizumoto et al.), however some more recent studies predict as high as 80% (Day M. “Covid-19: four fifths of cases are asymptomatic, China figures indicate.”). Mentions of such literature should be included.

We have added a citation in Table 1 of Oran and Topol (2020) on “Prevalence of Asymptomatic SARS-CoV-2 Infection” https://www.acpjournals.org/doi/pdf/10.7326/M20-3012 who reported that “Asymptomatic persons seem to account for approximately 40% to 45% of SARS-CoV-2 infections, and they can transmit the virus to others for an extended period, perhaps longer than 14 days”. We also included Baud et al. (2020), Mizumoto et al. (2020), Kirenga et al. (2020), Ing et al. (2020) and Nishiura et al. (2020) to highlight and emphasize the wide variation in reported asymptomatic percentages and we cite these four studies that are based on data in different countries. This is on page 18 in lines 372-375 as “… The other parameter that has been reported to vary widely across countries is the percentage of asymptomatic individuals. Several studies have estimated this proportion using data from different sources including hospitals and cruise ships and their estimated values have ranged from 18% to 81% [34-37]. …”

Similarly recovery rate, traced-and-isolated rates ( c ), infectivity in hospitals, need closer scrutiny. All model parameters need more source citations.

Limited data was somehow a hurdle to this analysis and we had to utilize all the then available information from Ministry of health addresses and press releases as well as data available in literature that was applicable to the Ugandan situation whenever possible. Where available, we have added new citations in Table 1 and also added some discussion points on the widely varying parameters.

The recovery rate was estimated from Ministry of information which indicated that an admitted patient spent 20 days in hospital on average. This information is in the same range with the data reported in the WHO- China report (see https://www.who.int/docs/default-source/coronaviruse/who-china-joint-mission-on-covid-19-final-report.pdf ) in which, using available preliminary data, the median time from onset to clinical recovery for mild cases was approximately 2 weeks and was 3-6 weeks for patients with severe or critical disease. We have added the WHO-China report as a secondary citation for the recovery rate in Table 1.

The traced-and-isolated rate (c) was set to 20% based on briefings from the Ministry of Health while addressing challenges in contact tracing due to its labor and other resources intensity as well as the misinformation and intentional refusal by the infected individuals to reveal all their contacts. This number could vary and in a scenario analysis, we assessed to impact of enhancing contact tracing and had (in the original manuscript) written that “…Finally the effect of enhancing contact tracing efforts is modeled through assuming that after 58 days, twice as many latently infected individuals are traced and all infectious periods are only one day…” However, now that we are trimming the content of the manuscript, that content is unfortunately omitted and only the sensitivity analysis has information on the potential impact of contact tracing magnitude on the outcomes. 

On infectivity in hospitals, note that Uganda took more than 2 months to register a COVID-19 case in frontline health workers. The first case among health workers was registered in the last week of May 2020 https://www.newvision.co.ug/news/1526183/covid-19-senior-epidemiologist-dies) and by the time of our analysis, the number of hospital-acquired infections was nearly negligible. However, we assessed the impact of having hospital acquired infections as part of the sensitivity analysis (still in the manuscript) and also in a scenario analysis (which has since been omitted following the recommendations). 

Parameter q can be removed if it is not used in the model.

This comment is appreciated but we thought that, for model flexibility, completeness and robustness, we should maintain the parameter q in the ode system. However, we have improved on the assumption to add more justification on page 8 in lines 155-158 as “… (h) Although viral loads have been reported to be similar between asymptomatic and symptomatic patients [30], the asymptomatically infected individuals may have a reduced infectivity because they may not cough or sneeze as much as the symptomatic. This however needs to be further elucidated. …”

Moving on to the model itself, a major concern is that individuals who are contacted via test-and-trace are moved to the hospitalisation class (If I understood correctly). This is a major oversight as it will be a significant move of individuals, having a considerable impact on the resulting dynamics, and invalidating any assessment of total hospitalisation numbers (such as figures 2,3,5). This estimation of hospitalised individuals is a major use of these models, and so the assessment must be rigorous.

Yes, you understood the transition correctly as was regrettably originally (poorly) written. 

We had missed a critical component describing the transition from E directly to H. We had not explicitly mentioned the fact that traced E individuals can only transit to H after completing their latent period. The proportion c is first isolated in hospital-like isolation settings until they test positive. By doing this, their infectious period is technically reduced to zero and are thus denied a chance of infecting others in the absence of hospital-acquired infections. 

We have now improved the write up on this assumption to capture this key concept on page 7 in lines 134-139 to “… (b) Upon infection, some latently infected individuals can be identified (e.g., through contact tracing) and subsequently individually isolated under high biosecurity conditions leading to their eventual hospitalization upon testing positive (i.e. completing their latent period), thereby denying them a chance of ever infecting other individuals in the community, while others may remain in the community and become infectious up until when they are identified and hospitalized. …”. 

Technically, from the transition term from E to H occurring at a rate cρE, it can be seen that the proportion c of the latently infected individuals still first go through their latent period before transiting to the H compartment and hence this transition has no obvious effect on the predicted number of hospitalized. For model parsimony, we did not include a separate isolation compartment since there was no intension to draw conclusions on the dynamics in the isolated compartment. However, even without that compartment, the alternative formulation is robust and it captures all the key processes and it carters for the latent period of the traced E individuals. Hence we believe the formulation used does not affect the dynamics in the H compartment. 

Practically, the transition from latently infected compartment directly to the hospitalized compartment as modeled was intended to mimic the reality on ground where, once identified as contacts, individuals were individually isolated in designated centers with no possibility of mingling with other (susceptible or otherwise) individuals besides health workers who are always protected with PPE. These individuals were regularly tested for COVID-19 and moved to treatment centers upon testing positive. This nature of isolation mimicked the settings in hospital in that the individuals in isolation that later turned out to have been latently infected by the time of their tracing did not get a chance to infect other individuals (outside hospital-acquired infections) upon becoming infectious. 

I am also sceptical of the implementation of imported cases. Having this influx of individuals (and I don’t fully understand how these imported cases subsequently leave the population pool in an equal amount) be dependent on the host population size (N) doesn’t make sense. In general, one could perhaps argue for a constant population (in/out) from the infected class, but this formulation is not suitable for then drawing conclusions on the dynamic impact of imported cases from neighbouring countries. It would be better to develop a model of multiple interacting SEIHR networks for this. I would recommend either removing this aspect or better framing this as “the dynamic impact of constant external disease pressure”.

On the implementation of imported cases as a function of host population size, we solely based this formulation on the fact that the demand for goods that the arriving truck drivers deliver would likely be directly proportional to the population size i.e. the demand and supply dynamics of economics. Since the timeframe over which the predictions were to be made was so short, we opted for the epidemic model in which we excluded the births and natural mortality. Over this short prediction period, we expect that, under the prevailing circumstances, the host population size may slightly fluctuate and hence dN/dt≠0.

On the suggestion to develop a model of multiple interacting SEIHR networks, we foresee a big hurdle relating to model parameterization and generally acquisition of outbreak data from Kenya, Tanzania (that officially already declared COVID-19 a hoax), South Sudan and Rwanda (that is under political tension with Uganda). The only readily available data was the number of arriving truck drivers (infected or not) that the Ministry of Health-Uganda releases daily based on daily testing at the border posts. It is likely that information derived from the testing data at the border posts may not be fully reflective of the disease dynamics in the neighboring countries. However, due to data limitations, we used that information in the current modeling framework.

The transient dynamics of the disease are happening simultaneously within the East African region. However, our model only incorporated the dynamism for the disease dynamics in Uganda and assumed a constant external disease pressure. We note that the coupling of the disease dynamics in Uganda with those in the neighboring through the arrival of infected truck drivers is partly governed by the disease prevalence in those countries. However, it is hard to get a grip on the required data from some of those countries to substantiate the network approach. 

Therefore, in line with your (greatly appreciated) technical advice on modeling, we have opted to reframe this particular analysis’ header on page 11 in line 228 and the section header for 4.3 on page 14 in line 283 as “… The dynamic impact of constant external disease pressure …”. We have also added a discussion point to this effect on page 18 in lines 376-380 as “… Given the demonstrated coupling of Uganda’s disease dynamics with those in her neighboring countries, network based modeling approaches would be better suited to assess potential impact of the coupling pathways on the disease dynamics. Although we assumed constant external disease pressure in this study, we recommend that future studies adopt the network modeling approach whenever relevant data to substantiate those network based models is available. …”

I would also be interested in more information on the exact lockdown rules in Uganda to defend the choice of lockdown modelling. The authors have a percentage of the population which is made impossible to become infected due to lockdown. This is appropriate for strict lockdown systems, such as that in Cyprus, where individuals may not leave home without permission, but for systems such as the UK where individuals may leave home freely for essential trips, a far reduced transmission rate may be more appropriate than a flat impossibility of transmission (e.g. Rawson et al. “How and when to end the COVID-19 lockdown”).

Our choice of lockdown modelling in which we assumed that a percentage of the population was impossible to become infected due to lockdown was motivated by level of strictness in implementing lockdown restrictions in Uganda which was one of the strictest in Africa. Content to this effect can be found in an article whose link is indicated from which I quote the following; “… In March, Uganda introduced one of the most stringent lockdowns in Africa, banning cars and public gatherings, shutting down shopping centres, places of worship, schools and entertainment centres, and putting in place a night-time curfew.” (See https://www.telegraph.co.uk/global-health/science-and-disease/ugandas-tough-approach-covid-19-hurting-citizens/ ). There were roadblocks manned by Military Police, the Army and Local defense personnel to ensure compliance. The measures were so strict to the extent of shooting some noncompliant citizens. Some of those acts are reflected in the BBC news article titled “Uganda - where security forces may be more deadly than coronavirus” (see https://www.bbc.com/news/world-africa-53450850 )

The approach of Rawson et al. 2020 in which the population was split into a quarantine and a non-quarantined group differing by the rate of virus transmission and the groups being connected by the modeled release strategies from lockdown seems to be one of the most suitable approaches for the countries in which lockdown restrictions were less strict.

We have added a justification for our choice of lockdown modeling in the discussion on page 18 in lines 380-386 as “On lockdown modeling, we assumed that a percentage of the susceptible population was isolated by the lockdown restrictions and hence impossible to become infected. This assumption is justified by the strictness in enforcing these measures in Uganda. However, for relatively lax implementation of the measures, alternative approaches such as that involving splitting the population into the quarantine and non-quarantine groups as done by [Rawson et al. 2020] could be more suitable.”

When the model is introduced (from L102), it is very difficult to interpret the model initially, needing to jump between table 1, figure 1, equations (1)-(5), and brief paragraphs. I would instead urge presenting each equation, one-by-one, and explaining specifically each term (and parameter) in each equation. This may seem excessive, but makes interpreting a model considerably easier.

The flow of content in this section has been improved up by numbering and then describing each equation fully in the text right below where all the equations appear. The content is capture on page 9 in lines 179-200 as “In Equation (3.1), the term (1-(A+E))ΛN represents the number of arriving susceptible individuals per day, ΒBS(QI_A+I_S+GH)/N represents the number of susceptible individuals that become latently infected per day, ΤR represents the number of previously immune individuals that lose their disease-induced immunity to become susceptible per day and ΛS represents the number of departing susceptible individuals per day. In Equation (3.2), the term EΛN represents the number of arriving latently infected individuals per day while ΡE represents the number of latently infected individuals that progress to become infectious per day. In Equation (3.3), the term AΛN represents the number of arriving asymptomatically infectious individuals per day, RΡE represents the number of latently infected individuals that become asymptomatically infectious per day and Ω_A I_A represents the number of asymptomatically infectious individuals that are hospitalized per day. In Equation (3.4), the term (1-(C+R))ΡE represents the number of latently infected individuals that become symptomatically infectious per day, Σ_S I_S represents the number of symptomatically infectious individuals that succumb to the disease before being hospitalized per day and Ω_S I_S represents the number of symptomatically infectious individuals that are hospitalized per day. In Equation (3.5), the term CΡE represents the number of latently infected individuals that were traced and isolated prior to being hospitalized upon testing positive per day, Ω_A I_A represents the number of asymptomatically infectious individuals that are hospitalized per day, Ω_S I_S represents the number of symptomatically infectious individuals that are hospitalized per day, Σ_H H represents the number of hospitalized individuals that succumb to the disease per day while ΑH represents the number of hospitalized individuals that recover from the disease per day. Lastly, in Equation (3.6), the term ΛR represents the number of departing recovered individuals per day and the other two terms are described above.”

The above issues then raise concern with the interpretation of results:

It is true that, if not well clarified as was the case in the earlier version of the manuscript, doubt would be cast on the interpretation of the results. However, we hope that the explanations given, amendments and clarifications made will eliminate (some of) the original doubt.

4.2 – This section seems the most powerful to focus on, and could be expanded by comparing and fitting to Ugandan case data. 

The model was fitted to Uganda’s case data for the first 58 days since the confirmation of the first case (when this analysis commenced) and the figure is below. The figure was helpful in assessing our model biological plausibility during parametrization.

The current primary result is not particularly useful or enlightening, less people in lockdown = more people infected. More interesting would be to then expand this to ask “how strict does a lockdown need to be to not exceed hospital capacity?” “When should a lockdown be ended?” etc. The current metric of hospitalisations is incorrect due to the above test-and-trace classification issues.

We agree that there could be other useful results from this study and we also acknowledge that we had not clearly explained how we implemented the trace-test-and- hospitalize in latently infected class, and this could have partly led to your doubting of the hospitalization metric. We have now addressed the latter while addressing your previous comment above and also added more content on this at the end of the rebuttal to this very point. We hope that the amendments and explanations given on this matter suffice.

One of the primary aims of the study was to inform policy on how to release lockdown restrictions i.e., under a given percentage of susceptible individuals, when will the then 3200 hospital capacity. On this, we report on page 14 lines 278-281 as “… These results demonstrate that if we ease lockdown by releasing 50% of susceptible population for the Ugandan situation with 3200 beds and not all are of ICU-like capacity, within 100 days the COVID-19 related hospitalization demand would have already overwhelmed the current resources. …” This is in line with your opinion that we frame the study to address the question; “how strict does a lockdown need to be to not exceed hospital capacity?”. 

Additionally, on using the study to answer the question; “When should a lockdown be ended?”, we address a closely related question in section 4.4 on the possibility of a second wave on page 15 in line 297. In that analysis, we determine when the government can let up to 75% of the population out of lockdown while mitigating the second wave possibility. We report on page 15 in lines 302-304 as “… The simulations reveal that with the current levels of contact tracing, having released up to 75% of susceptible after 150 days would guarantee emergence of a second epidemic wave. …” and also in Figure 4 caption on page 27 lines 547-551 “… Panel (a) shows easing of lockdown from 10% to 30% susceptible after 58 days and further easing to 75% after 210 days of lockdown and could be sooner if contact tracing efforts were intensified. Panel (b) is a case of easing lockdown from 10% to 30% after 58 days followed by 75% susceptible after 150 days of lockdown – a case of too many and too soon. …”

On the point of a possible incorrectness of the hospitalization metric that your raise, we think that that is rectified by the fact that the traced and tested latently infected individuals will on join the Hospitalized compartment upon completing their latent period (since there are first individually isolated until testing positive after which they are taken to hospital). We believe that this alternative formulation still captures the dynamics appropriately since it does not in any way reduce on the latent period. The only difference that this alternative formulation correctly captures is the elimination of the infectious potential for the traced and isolated latently infected individuals since they are not accorded a chance to mix with the susceptible individuals upon becoming infectious; they will be hospitalized straightaway. 

4.3 – I currently don’t think this model formulation is powerful enough to answer questions on the impact of imported cases. The result that “COVID remains endemic” is due precisely to the model formulation. The constant influx of infection from these cases ensures an endemic outbreak. If the authors wanted to investigate the impact of imported cases then this influx should be variable. Then the model can answer questions such as “When should imports from country A be allowed based on their case rates?”.

Agreed! 

We have eliminated the subsection detailing the analyses of “The impact of hospital acquired infections, regional disease prevalence and contact tracing efforts”. Former subsection 4.5 and Figure 4.5 have been omitted from the revised manuscript. We also revised the results and discussion accordingly. The results on potential impact of varying contact tracing effort are now only discussed in line with the sensitivity analysis. For example, in the results on page 15 in lines 314-319 as “… From the sensitivity analysis, the most impactful parameters identified include the arriving proportion of infected individuals (a), the per capita recruitment rate (λ), the disease transmission rate (β), the proportion of susceptible population available for infection (b), and the efforts of contact tracing and subsequent isolation of latently infected individuals (c) among others. …” Otherwise, these parameters only appear at their default values in the other analyses. The potential impact of varying hospital acquired infections is now discussed on page 18 in lines 360-365 as “… Although the parameter influencing the number of hospital acquired infections was of relatively low impact on the number of symptomatic cases in the sensitivity analysis, field experience with handling COVID-19 in other countries (note that Uganda did not report any hospital acquired infection for the first two months of her epidemic) as well as diseases such as Ebola in Uganda dictates that this should be maintained at a bare minimum to ensure that health workers do not shun their duties. …” 

4.4 – This section will be insightful, however the above concerns on model formulation and parameterisation need to be addressed first.

Our original shallowness in explaining some of the key approaches and assumptions (such as that on the trace-test-and-Hospitalize a fraction of the latently infected individuals) in our model could have led to most of the queries raised. We have improved the write up as described under each of queries raised above. We hope that all or most of the concerns have now been addressed and we also hope that the amendments and rebuttals are helpful in generating more insights on our approaches. 

4.5 – I don’t think the model is suitable for assessing the impact of hospital-acquired infections currently. This is an interesting dynamic, but would benefit from a more bespoke model, with separate healthcare compartmentalisations. The current hopsitalisation framework (see above) invalidates the current results.

Following your recommendation, we have eliminated this subsection and refocused the manuscript accordingly. The originally assessed parameters are now only mentioned in the sensitivity analysis results e.g., on page 15 in lines 314-318 as “… From the sensitivity analysis, the most impactful parameters identified include the arriving proportion of infected individuals (a), the per capita recruitment rate (λ), the disease transmission rate (β), the proportion of susceptible population available for infection (b) and the efforts of contact tracing and subsequent isolation of latently infected individuals (c) among others. …”

4.6 – I’m not sure the text here actually correlates with the sensitivity analysis of Figure 6. The text states that a, e, b, and g are most influential. However Figure 6 seems to infer that a and lambda are the most impactful, and the others have minor impact. Certainly g and e seem of low impact. Also, the bins of Figure 6 should be marked with their respective parameters. The current colour scheme is hard to distinguish.

This error has been rectified on page 15 in lines 314-318 as “… From the sensitivity analysis, the most impactful parameters identified include the arriving proportion of infected individuals (a), the per capita recruitment rate (λ), the disease transmission rate (β), proportion of susceptible population available for infection (b) and the efforts of contact tracing and subsequent isolation of latently infected individuals (c) among others. …”.

The Figure (now #5) identifies the parameters using the legend on the right in the order of appearance of the bars in the graph with the topmost entry in the legend corresponding to the first bar in the graph. The figure caption on page 28 in lines 556-562 has be improved to explicitly explain this as “…Figure 5. Pearson’s Rank Correlation Coefficients (PRCCs) tornado simulation of the model’s most sensitive parameters sampled 300 times using the Latin hypercube sampling scheme. Sensitivity variation with respect to time is analyzed at two time points, 30 days and 58 days that could be used as initialization points or start times for the intervention measures. The order of appearance of parameters in the figure legend follow the left to right order in labeling the corresponding bars. For example, the topmost label corresponds to the first bar in the figure and so on. …” 

Minor Comments:

L51/L52 – “measures reduced the susceptible population to about 10%”, this is important, is there a citation for this?

This was mentioned in press briefings by the Ministry of Health and other non-documented outlets. However, it can be approximated based on the work demographic data of Uganda (Source: Uganda Bureau of Statistics) where 80% of Ugandan’s workforce is employed in the non-formal sector and hence were less likely to have been categorized as “Essential workers” as per the definition used during lockdown. Only a few market vendors (who were not even allowed to return home to their families) as well as essential, health service providers and essential cargo transporters were allowed to operate. Also of the 43 million Ugandans, 15 million are school going and hence congregate in schools yet all schools were closed during lockdown. 

L129 – Does this mean that asymptomatic individuals are hospitalised? I assume this is due to the test-and-trace, this needs to be changed.

Yes. Everyone that tested positive- symptomatic or not- had to be hospitalized to prevent them from spreading the disease in the community. This went on until recently (as recent as early November 2020) when the hospitals got overwhelmed.

L266 – What is meant by undetected infectious here? Mentioned specifically which model variable you mean when referenced (e.g. I/E)

The undetected infectious individuals referred to there and throughout the manuscript comprise of the yet-to-be hospitalized asymptomatic and sympotomatic infectious individuals. We have explicitly defined this concept on its first appearance in the manuscript on page 14 in lines 277-278 as “…the undetected infectious individuals (comprising of the yet-to-be hospitalized asymptomatic and symptomatic infectious individuals)…”

The model is good and I think it is of critical importance to investigate country-specific disease trajectories, the scope and details of the model just need to be refined. I look forward to seeing a more refined revision and believe it will make a strong contribution when finished.

This encouraging comment as well as the suggested amendments are greatly appreciated.

---

## [Decision Letter · Decision Letter 2]

4 Jan 2021

PONE-D-20-24477R2

Mathematical modeling of COVID-19 transmission dynamics in Uganda: Implications of Complacency and Early Easing of Lockdown

PLOS ONE

Dear Dr. Ssematimba,

Thank you for submitting your manuscript to PLOS ONE. After careful consideration, we feel that it has merit but does not fully meet PLOS ONE’s publication criteria as it currently stands. Therefore, we invite you to submit a revised version of the manuscript that addresses the points raised during the review process.

These issues are minor, and after they are addressed I expect we will be able to accept the manuscript for publication.

We look forward to receiving your revised manuscript.

Kind regards,

Siew Ann Cheong, Ph.D.

Academic Editor

PLOS ONE

Reviewers' comments:

Reviewer's Responses to Questions

**Comments to the Author**

1. If the authors have adequately addressed your comments raised in a previous round of review and you feel that this manuscript is now acceptable for publication, you may indicate that here to bypass the “Comments to the Author” section, enter your conflict of interest statement in the “Confidential to Editor” section, and submit your "Accept" recommendation.

Reviewer #1: All comments have been addressed

Reviewer #3: (No Response)

2. Is the manuscript technically sound, and do the data support the conclusions?

Reviewer #1: Yes

Reviewer #3: Partly

3. Has the statistical analysis been performed appropriately and rigorously? 

Reviewer #1: Yes

Reviewer #3: Yes

4. Have the authors made all data underlying the findings in their manuscript fully available?

Reviewer #1: Yes

Reviewer #3: Yes

5. Is the manuscript presented in an intelligible fashion and written in standard English?

Reviewer #1: Yes

Reviewer #3: Yes

6. Review Comments to the Author

Reviewer #1: Authors are precisly addressed the comments by adding comprehensive discussion in article. I recommend to accept the article in present form

Reviewer #3: See attached document for full comments.

In summary, the final revision needed is just some final justification and explanation of some model parameters in the manuscript. Mortality rate will likely need to be changed and, as such, figure plots may need to be re-done.

"

SUMMARY

The revisions made address almost all of my points very well. There is only one point that needs addressing before I can finally recommend for publication, and that is regarding the model parameter values again. This work will be extremely useful in the coming years I believe for investigating the impact of different mitigation strategies, and as such it is important to make sure that these parameter values are well cited and explained for future readers to understand the measures unique to Uganda.

Essentially I would just want to see some more information for those parameters that cite reference [4]. Naturally, some of these parameters will be deeply uncertain, and so “back-of-the-envelope” estimates will need to be used. There is nothing wrong with this, however I would want to see exactly how these values were reached. As this could involve a lot of text, I would perhaps recommend moving these calculations to a separate appendix or supplementary info page.

Also, where specific press-releases or ministerial briefings are used, these specific briefings should be cited, such as the ones that the authors linked to me in the above comment regarding influx from other countries / lockdown amounts. In short, a lot of the justification given to me in responses by the authors should be included for all readers!

Really the main sticking point here is still the mortality rate. These estimates were based on old data that is clearly now not an accurate representation of the true mortality rate to my eyes. Indeed, based on today’s case data the total mortality rate is 0.007 in Uganda for confirmed cases. The current parameter value used is likely to really affect the results shown, which is important given some of the findings regarding hospital capacity. I think these really need to be re-run with an improved mortality rate, unless my understanding is incorrect?

Once this is addressed I will be happy to recommend for publication.

A minor point, I think it would be extremely helpful for reviewers if a paragraph was included in the Introduction that outlines the specific, extreme, protocols used in Uganda. This motivates the work extremely well, showing why bespoke modelling frameworks are needed.

Finally,

L295 – “Even more worrying is the fact that the endemic level would higher at bigger levels of susceptibility” – I think there’s a typo here, I’m a bit confused about what is trying to be said with this sentence?

"

7. PLOS authors have the option to publish the peer review history of their article (what does this mean?). If published, this will include your full peer review and any attached files.

Reviewer #1: No

Reviewer #3: **Yes: **Thomas Rawson

---

## [Author Response · Author response to Decision Letter 2]

20 Jan 2021

20 January 2021

Revision of manuscript number PONE-D-20-24477R2 titled “Mathematical modeling of COVID-19 transmission dynamics in Uganda: Implications of Complacency and Early Easing of Lockdown” 

On behalf of the co-authors, we appreciate all your efforts devoted towards improving this manuscript. Thank you so much!

Initial Reviewer Comments

Author Responses

Reviewer Responses

Author Responses to the new queries

Reviewer #3: The manuscript utilises a compartmental SEIHR framework to assess the disease dynamics in Uganda, and consider how to tackle future infection possibilities. Current manuscript revisions have already improved the manuscript well. While I feel the work has great potential to advance the understanding of disease dynamics within Uganda, I have substantial concerns with the current model formulation, and the conclusions the authors draw.

The primary issue is that the authors attempt to, arguably, do too much with the model, and in doing so each conclusion suffers. I list below my general concerns with the model formulation, and which scenarios are ill-posed for the current model. In general I would urge the authors to restrict their scope, and tweak the model to draw stronger conclusions on a specific research question or two. I feel the model may be best suited with minor adjustment to assessing “The possibility of a second wave infection”, or “The impact of lockdown measures”, while it is currently not an appropriate model for assessing “The impact of magnitude of imported cases” or “Impact of hospital acquired infections”.

The Introduction is very strong and suitably frames the work.

We agree that trimming and refocusing the content tremendously improves that quality of the manuscript and as suggested by the reviewer, the manuscript has been refocused to address three specific questions namely; 1) assessing the potential impact of the implemented lockdown measures, 2) the dynamic impact of constant external disease pressure (phrasing suggested by the reviewer in the later comments), and 3) the possibility of the second wave of infection. We also maintain the section on sensitivity analysis.

I thank the authors for the time taken to address my concerns. The manuscript is hugely improved and presents a very strong and interesting model investigating the specific mitigation strategies and disease-trajectory within Uganda. I must however request some minor additional adjustments before I can finally recommend for publication, primarily regarding the justification of some model parameters. Justification for these is discussed in the below responses of the authors, however this needs to also be conveyed in the manuscript itself. I address these specific points directly below, and summarise at the end. Apologies to further slow the process but these model parameters must be well-explained for readers, I appreciate this may require a re-plotting of some figures and results presentation, which is always frustrating, but I think it is very important for the final presentation of results.

We greatly appreciate your commitment and devotion to improving the quality and eventual impact and usability of this manuscript. As you rightly foresaw, we had to rerun the simulations and replot all the graphs. Luckily, the resulting changes in predictions were mostly non-fundamental with only a shift in days of epidemic peaking. Importantly and perhaps expectedly, the model’s dynamical behaviour is mostly unchanged. All in all, the task at hand was worthwhile since the model is now, among others, parameterized using updated disease-induced mortality and hospital-acquired infection rates that reflect the current epidemic trend in Uganda thus making the predictions more reliable for field adoption. 

The key changes include a shift in the timeframe required to completely eliminate the disease to nine months as depict in the new Figure 3 and also captured in the abstract on page 2 in lines28-29 as “The study findings show that even with elimination of all imported cases at any given time it would take up to nine months to rid Uganda of the disease. ….”, in the discussion on page 17 in lines 345-347 as “… In all the assessed scenarios the disease would be wiped out in the case where there are no infected arrivals beyond the first 58 days and in this case the disease would be wiped out within 270 days as seen from Figure 3…”. Another key shift was in the appropriate time to release 75% of the population from lockdown. This shift is depict in Figure 4 and are also captured on page 16 lines 318-320 as “…The simulations reveal that with the current levels of contact tracing, having released up to 75% of susceptible after 210 days would guarantee emergence of a second epidemic wave. …” and in the discussion on page 18 in lines 363-368 as “…The results in Figure 4 reveal that, even in a fairly ideal situation with no new arrivals of imported cases, once the lockdown is hurriedly lifted to a 75% level, the yet-to-be detected cases in the community have potential to start a second and more disastrous epidemic wave. Note however that with enhanced surveillance and contact tracing, gradual easing by releasing smaller percentages of susceptible individuals from lockdown can still be safely executed sooner than the optimum 300 days for the 75%. …”

General Concerns

My initial concerns are with the model parameters (Table 1). Many parameters cite the Ugandan data portal (reference [4]) for their values, however I cannot locate these model parameters, nor can I infer them from Ugandan case data.

For reliability of the study outcomes in informing Uganda policy interventions to manage the disease, it was necessary that the study uses localized parameters whenever possible. However, as was the case with other earlier modeling studies, only scanty information about the pathogen and the disease in general was available and more is just being generated as the pandemic progresses. Therefore, some of the earlier simulation studies utilized the limited data together with sensitivity analyses to infer disease dynamics under varying intervention scenarios. It is hoped that as more knowledge about the pathogen and its transmission pathways in generated, the models will be updated accordingly.

For our Ugandan case, not all required parameters have been documented in citable literature and this was a hurdle to the current study. For example, besides the number of new cases, recoveries and deaths that are mandatorily documented and forwarded to WHO among other stakeholders, the other parameters that cite the Ministry of Health data (reference 4) are mainly inferred or derived from the routine addresses by the personnel from the Ministry of Health, Uganda virus research institute and the office of the presidency among others and are based on observational data. Examples of such include the arrival rate of truck drivers (both infected and non-infected), number of returning residents through airports, time spent in hospital for the admitted cases which partly informs the model recovery rate, the proportion of asymptomatic infected individuals etc. In our parameter estimation, content from these routine national addresses and press releases which is often summarized in news outlets was used to infer some of the model parameters. One of such briefings can be found online at https://www.health.go.ug/document/update-on-covid-19-response-in-uganda/ and https://www.health.go.ug/covid/category/press-release/ Studies are being conducted and it is hoped that such new studies will have data from which other more precise parameters can be obtained. For example the recently published Kirenga et al. (2020) “Characteristics and outcomes of admitted patients infected with SARSCoV-2 in Uganda” https://www.ncbi.nlm.nih.gov/pmc/articles/PMC7477797/pdf/bmjresp-2020-000646.pdf validates some of the parameter estimates.

We have checked the literature again and identified some articles that we could cite for some of the parameters in Table 1. For recruitment rate [Ministry of Health -Uganda. UPDATE ON THE COVID-19 OUTBREAK IN UGANDA available online at https://www.health.go.ug/covid/category/press-release/ accessed on 1 December 2020. 2020.], for recovery rate of hospitalized individuals [WHO-China. Report of the WHO-China Joint Mission on Coronavirus Disease 2019, available online at https://www.who.int/docs/default-source/coronaviruse/who-china-joint-mission-on-covid-19-final-report.pdf, accessed 3 June 20202020.] and for percentage of latently infected individuals in community that becomes asymptomatic [Daniel P. Oran A, Eric J. Topol. Prevalence of Asymptomatic SARS-CoV-2 Infection. Annals of Internal Medicine. 2020;173(5):362-7. doi: 10.7326/m20-3012 %m 32491919.]

These are good additions. I of course appreciate the difficulties in ascertaining specific model parameters, and do not take issue with “back-of-the-envelope” calculations from case data. Rather it just needs to be made clear which parameters are rougher estimates and which are not.

It is a huge relief and reassurance to know that the reviewer acknowledges and appreciates the difficulties in parametrizing mathematical models for novel pathogens. We have improved on the traceability of reported parameter sources by citing the accessible online sources whenever available as well as Personal communications on reported but undocumented/inaccessible content. 

For the parameters that were derived from the televised press briefings by the Ministry of health official and COVID-19 task force in Uganda (that could not be accessed online), we have revised their sources by adding footnotes to describe how we dealt with the televised communications that had information pertaining to them. For example, the recovery rate of hospitalized individuals was estimated from the reported but yet-to-be documented average time spent in hospital. Also, the hospital acquired infection rate is also obtained from such an address. For purposes of credible citation of information source, we sought expert opinion and re-affirmation in some cases from the Uganda COVID-19 task force members and Ministry of health personnel on the COVID-19 frontline as well as distinguished economists following the standard procedures of obtaining and citing personal communications. We now use the expert-derived information directly and cite the corresponding personal communications. 

Some parameters are also very far from similar parameter estimates in the literature. For example, a disease-induced mortality rate of 0.001 is far low than estimates elsewhere in the literature. The hospital mortality rate of 0.0001 is even more peculiar. See for example, Baud et al. (2020) “Real estimates of mortality following COVID-19 infection.”.

The low disease-induced death rate that used in our study is for the Ugandan situation where Uganda did not register any COVID-19 related death in the first four months from the first case (i.e. March 22nd to July 22nd 2020). The first death was registered on 23rd July 2020 and at that point, the cumulative number of cases was 1079 with 971 recoveries. In other countries, using data as of March 1st 2020 (i.e. approx. 4 months (December to March) into the outbreak), Baud et al. (2020) reported mortality rate of 3·6% (95% CI 3·5–3·7) in China and 1·5% [1·2–1·7] outside of China and when adjusted for incubation period delay, mortality rates would be 5·6% (95% CI 5·4–5·8) for China and 15·2% (12·5–17·9) outside of China. Moreover, even as of December 1st 2020, globally there are approximately 63.7m cases, 1.5m deaths and 44.1m recoveries while in Uganda, there have been 20,459 cases, with 205 fatalities and 8989 recoveries. These numbers may crudely indicate both the case-fatality ratio and Recovered-fatality ratio (obtained from considering only the resolved cases) are relatively lower in Uganda.

For completeness and clarity, we have added the following “justification” also citing Baud et al (2020) in the discussion on page 18 in lines 367-372 as “… There are some limitations to our analysis that may either arise from the assumptions of the model or its parametrization since we used biologically plausible parameters based on current evidence yet some may be refined as more comprehensive data become available. For example, the COVID-19 induced mortality rate derived for this study based on data from Uganda is relatively lower than global values (e.g., those reported in [33]) and this may be validated in future. …”

Following the quoted numbers of 205 deaths and 20,459 cases, this equates to a mortality rate of ~0.01, as opposed to either 0.001 or 0.0001 as used in the manuscript. I understand then that the authors have based their mortality rates on the 1 out of 1079 cases from the initial outbreak period. I think it is however clear to see as testing has increased that this initial amount was not indicative of the true mortality rate. Some parameter inaccuracy is of course permissible given how much is still unknown, and I of course encourage using data specific to Uganda, however the values used show signs of being out by a factor of 10 or 100. This will also directly impact the number of hospitalisations observed. I am comfortable with a general difference in mortality between hospitalised and non-hospitalised, however 0.0001 is just a bit too far to my mind! In general, something like multiplying both these factors by 10 (0.01 and 0.001 respectively) seems a more sensible bet, and closer to the true mortality rate witnessed in Uganda. 

Following the reviewer’s suggestion, which we greatly appreciate, we have updated the disease-induced mortality rate to 0.008 based on the reported cases and deaths in Uganda as of December 28th 2020. For this we now cite “Ministry of Health -Uganda. COVID-19 RESPONSE INFO HUB STATISTICS. Last updated December 28, 2020. Available online at https://covid19.gou.go.ug/statistics.html Accessed on January 6, 2021 2021.” We also introduce a factor of 10% to have a rate of 0.0008 among the hospitalized as proposed by the reviewer.

An estimate of 40% asymptotic individuals could hold true, there are multiple studies citing closer to 20% (e.g. Mizumoto et al.), however some more recent studies predict as high as 80% (Day M. “Covid-19: four fifths of cases are asymptomatic, China figures indicate.”). Mentions of such literature should be included.

We have added a citation in Table 1 of Oran and Topol (2020) on “Prevalence of Asymptomatic SARS-CoV-2 Infection” https://www.acpjournals.org/doi/pdf/10.7326/M20-3012 who reported that “Asymptomatic persons seem to account for approximately 40% to 45% of SARS-CoV-2 infections, and they can transmit the virus to others for an extended period, perhaps longer than 14 days”. We also included Baud et al. (2020), Mizumoto et al. (2020), Kirenga et al. (2020), Ing et al. (2020) and Nishiura et al. (2020) to highlight and emphasize the wide variation in reported asymptomatic percentages and we cite these four studies that are based on data in different countries. This is on page 18 in lines 372-375 as “… The other parameter that has been reported to vary widely across countries is the percentage of asymptomatic individuals. Several studies have estimated this proportion using data from different sources including hospitals and cruise ships and their estimated values have ranged from 18% to 81% [34-37]. …”

Similarly recovery rate, traced-and-isolated rates ( c ), infectivity in hospitals, need closer scrutiny. All model parameters need more source citations.

Limited data was somehow a hurdle to this analysis and we had to utilize all the then available information from Ministry of health addresses and press releases as well as data available in literature that was applicable to the Ugandan situation whenever possible. Where available, we have added new citations in Table 1 and also added some discussion points on the widely varying parameters.

The recovery rate was estimated from Ministry of information which indicated that an admitted patient spent 20 days in hospital on average. This information is in the same range with the data reported in the WHO- China report (see https://www.who.int/docs/default-source/coronaviruse/who-china-joint-mission-on-covid-19-final-report.pdf ) in which, using available preliminary data, the median time from onset to clinical recovery for mild cases was approximately 2 weeks and was 3-6 weeks for patients with severe or critical disease. We have added the WHO-China report as a secondary citation for the recovery rate in Table 1.

The traced-and-isolated rate (c) was set to 20% based on briefings from the Ministry of Health while addressing challenges in contact tracing due to its labor and other resources intensity as well as the misinformation and intentional refusal by the infected individuals to reveal all their contacts. This number could vary and in a scenario analysis, we assessed to impact of enhancing contact tracing and had (in the original manuscript) written that “…Finally the effect of enhancing contact tracing efforts is modeled through assuming that after 58 days, twice as many latently infected individuals are traced and all infectious periods are only one day…” However, now that we are trimming the content of the manuscript, that content is unfortunately omitted and only the sensitivity analysis has information on the potential impact of contact tracing magnitude on the outcomes.

On infectivity in hospitals, note that Uganda took more than 2 months to register a COVID-19 case in frontline health workers. The first case among health workers was registered in the last week of May 2020 https://www.newvision.co.ug/news/1526183/covid-19-senior-epidemiologist-dies) and by the time of our analysis, the number of hospital-acquired infections was nearly negligible. However, we assessed the impact of having hospital acquired infections as part of the sensitivity analysis (still in the manuscript) and also in a scenario analysis (which has since been omitted following the recommendations).

These are good justifications, and as such need to be included in the manuscript! I appreciate this is quite a lot of text to include, so I think my best advice would be to create a separate appendix or supplementary file that directly explains the inference of all these model parameters from case data and ministerial reports / press-releases.

For the parameters that could not be directly derived from the cited literature, we have added brief descriptions on how they were derived for the cited sources in the footnotes in Table 1. We have also added on page 9 in lines 182-184 as “… For some of the parameter values derived from the Ministry of Health press briefs and other government agency websites, we describe what information was utilized in their respective footnotes in Table 1. …”. Below are the parameter specific details.

The recruitment rate λ was estimated based on Ministry press releases in which between 1000 and 2000 drivers were reportedly arriving and being tested. We now indicate it as “Estimated” and mention how this was achieved in the footnote in Table 1 page 31 in line 608 as “… aEstimated based on testing data of arriving truck drivers that was between 1000 and 2000 per day.…”

The fraction susceptible (parameter b) which captures the percentage of people that continued to move freely during the initial phase of the lockdown was estimated from data on employment by sector from the Uganda Bureau Statistics (UBOS) data, number of school goers together with the information on how COVID-19 control measures were strictly implemented in Uganda. We now indicate it as “Estimated” and mention how this was achieved in the footnote in Table 1 on page 31 in lines 609-611 as ”… bEstimated based on the Uganda Bureau of Statistics data that of the 43 million Ugandans, 15 million are school going, >75% of adults engaged in informal sector and the strictness of lockdown measures the initially left only essential workers of approximately 4 million to mingle….”

” .

The infectivity of hospitalized individuals was merely assumed and we now indicate it as “Assumed” and mention how this was achieved in the footnote in Table 1 on page 31 in lines 6012-631 as “… cAssumed based on the report that 530 health workers had been infected with 6 fatalities by 30 September 2020….”

The traced-and-isolated rate (parameter c) was estimated based on the number of people that were testing positive while in quarantine that was reported by the Ministry of Health-Uganda. We now indicate it as “Estimated” and also capture this in the footnotes of Table 1 on page 31 in lines 614-615 as” … dEstimated based on the reported number of new infections reported among isolated individuals in MoH press briefings… .” 

Parameter q can be removed if it is not used in the model.

This comment is appreciated but we thought that, for model flexibility, completeness and robustness, we should maintain the parameter q in the ode system. However, we have improved on the assumption to add more justification on page 8 in lines 155-158 as “… (h) Although viral loads have been reported to be similar between asymptomatic and symptomatic patients [30], the asymptomatically infected individuals may have a reduced infectivity because they may not cough or sneeze as much as the symptomatic. This however needs to be further elucidated. …”

This is a good idea.

Moving on to the model itself, a major concern is that individuals who are contacted via test-and-trace are moved to the hospitalisation class (If I understood correctly). This is a major oversight as it will be a significant move of individuals, having a considerable impact on the resulting dynamics, and invalidating any assessment of total hospitalisation numbers (such as figures 2,3,5). This estimation of hospitalised individuals is a major use of these models, and so the assessment must be rigorous.

Yes, you understood the transition correctly as was regrettably originally (poorly) written.

We had missed a critical component describing the transition from E directly to H. We had not explicitly mentioned the fact that traced E individuals can only transit to H after completing their latent period. The proportion c is first isolated in hospital-like isolation settings until they test positive. By doing this, their infectious period is technically reduced to zero and are thus denied a chance of infecting others in the absence of hospital-acquired infections.

We have now improved the write up on this assumption to capture this key concept on page 7 in lines 134-139 to “… (b) Upon infection, some latently infected individuals can be identified (e.g., through contact tracing) and subsequently individually isolated under high biosecurity conditions leading to their eventual hospitalization upon testing positive (i.e. completing their latent period), thereby denying them a chance of ever infecting other individuals in the community, while others may remain in the community and become infectious up until when they are identified and hospitalized. …”.

Technically, from the transition term from E to H occurring at a rate cρE, it can be seen that the proportion c of the latently infected individuals still first go through their latent period before transiting to the H compartment and hence this transition has no obvious effect on the predicted number of hospitalized. For model parsimony, we did not include a separate isolation compartment since there was no intension to draw conclusions on the dynamics in the isolated compartment. However, even without that compartment, the alternative formulation is robust and it captures all the key processes and it carters for the latent period of the traced E individuals. Hence we believe the formulation used does not affect the dynamics in the H compartment.

Practically, the transition from latently infected compartment directly to the hospitalized compartment as modeled was intended to mimic the reality on ground where, once identified as contacts, individuals were individually isolated in designated centers with no possibility of mingling with other (susceptible or otherwise) individuals besides health workers who are always protected with PPE. These individuals were regularly tested for COVID-19 and moved to treatment centers upon testing positive. This nature of isolation mimicked the settings in hospital in that the individuals in isolation that later turned out to have been latently infected by the time of their tracing did not get a chance to infect other individuals (outside hospital-acquired infections) upon becoming infectious.

My apologies to the authors, as I was not aware of the specific isolation protocol within Uganda. Indeed, having learnt of this, it strengthens my belief that this work will make a strong contribution as a study of country-specific protocol. I might even recommend adding a brief paragraph to the introduction explicitly stating the use and protocol of isolation centres, not just to defend the modelling choices, but to stress the importance of this work. I found this document to be very useful in understanding the procedures: https://www.health.go.ug/covid/wp-content/uploads/2020/04/Updated-Guide-Draft_24April20_CLEAN.pdf and would recommend perhaps citing this to help explain to readers as naïve as I was! I might also suggest adding somewhere in the abstract how this model captures “specific isolation protocol followed in Uganda”, as I think this is a major strength of the model that should be advertised.

We are to blame for initially not clearly highlighting the specific isolation protocol in Uganda and that was almost costing us. 

Following your recommendation, we have added a sentence in the introduction on page 5 lines 91-96 as “…Country-variations in the implementation of COVID-19 control measures may require formulation and parametrization of country-specific mathematical models. It is noteworthy that the government of Uganda initially implemented one of the strictest control measures in Africa that, among others, included institutional isolation of all arriving individuals, contact persons of confirmed cases and a strictly enforced lockdown of all non-essential activities[23]….”

We have also added a sentence in the abstract as suggested on page 2 lines 24-27 as “…Novel in this model was the unique aspect of modeling the trace-and-isolate protocol in which some of the latently infected individuals tested positive while in strict isolation centers thereby reducing their infectious period….” and also in the introduction on page 5 in lines 98-100 as “The model was uniquely formulated in a such a way that it captures the trace-and-isolate protocol that was strictly implemented in Uganda.”

I am also sceptical of the implementation of imported cases. Having this influx of individuals (and I don’t fully understand how these imported cases subsequently leave the population pool in an equal amount) be dependent on the host population size (N) doesn’t make sense. In general, one could perhaps argue for a constant population (in/out) from the infected class, but this formulation is not suitable for then drawing conclusions on the dynamic impact of imported cases from neighbouring countries. It would be better to develop a model of multiple interacting SEIHR networks for this. I would recommend either removing this aspect or better framing this as “the dynamic impact of constant external disease pressure”.

On the implementation of imported cases as a function of host population size, we solely based this formulation on the fact that the demand for goods that the arriving truck drivers deliver would likely be directly proportional to the population size i.e. the demand and supply dynamics of economics. Since the timeframe over which the predictions were to be made was so short, we opted for the epidemic model in which we excluded the births and natural mortality. Over this short prediction period, we expect that, under the prevailing circumstances, the host population size may slightly fluctuate and hence dN/dt≠0.

On the suggestion to develop a model of multiple interacting SEIHR networks, we foresee a big hurdle relating to model parameterization and generally acquisition of outbreak data from Kenya, Tanzania (that officially already declared COVID-19 a hoax), South Sudan and Rwanda (that is under political tension with Uganda). The only readily available data was the number of arriving truck drivers (infected or not) that the Ministry of Health-Uganda releases daily based on daily testing at the border posts. It is likely that information derived from the testing data at the border posts may not be fully reflective of the disease dynamics in the neighboring countries. However, due to data limitations, we used that information in the current modeling framework.

The transient dynamics of the disease are happening simultaneously within the East African region. However, our model only incorporated the dynamism for the disease dynamics in Uganda and assumed a constant external disease pressure. We note that the coupling of the disease dynamics in Uganda with those in the neighboring through the arrival of infected truck drivers is partly governed by the disease prevalence in those countries. However, it is hard to get a grip on the required data from some of those countries to substantiate the network approach.

Therefore, in line with your (greatly appreciated) technical advice on modeling, we have opted to reframe this particular analysis’ header on page 11 in line 228 and the section header for 4.3 on page 14 in line 283 as “… The dynamic impact of constant external disease pressure …”. We have also added a discussion point to this effect on page 18 in lines 376-380 as “… Given the demonstrated coupling of Uganda’s disease dynamics with those in her neighboring countries, network based modeling approaches would be better suited to assess potential impact of the coupling pathways on the disease dynamics. Although we assumed constant external disease pressure in this study, we recommend that future studies adopt the network modeling approach whenever relevant data to substantiate those network based models is available. …”

I’m still a little uncertain as to the accuracy of how in/out-flux of travel from other countries is modelled. However this should not impact the results, so I’m happy for it to be included as it currently is.

This consideration is greatly appreciated is this is still a complex process to untangle. 

I would also be interested in more information on the exact lockdown rules in Uganda to defend the choice of lockdown modelling. The authors have a percentage of the population which is made impossible to become infected due to lockdown. This is appropriate for strict lockdown systems, such as that in Cyprus, where individuals may not leave home without permission, but for systems such as the UK where individuals may leave home freely for essential trips, a far reduced transmission rate may be more appropriate than a flat impossibility of transmission (e.g. Rawson et al. “How and when to end the COVID-19 lockdown”).

Our choice of lockdown modelling in which we assumed that a percentage of the population was impossible to become infected due to lockdown was motivated by level of strictness in implementing lockdown restrictions in Uganda which was one of the strictest in Africa. Content to this effect can be found in an article whose link is indicated from which I quote the following; “… In March, Uganda introduced one of the most stringent lockdowns in Africa, banning cars and public gatherings, shutting down shopping centres, places of worship, schools and entertainment centres, and putting in place a night-time curfew.” (See https://www.telegraph.co.uk/global-health/science-and-disease/ugandas-tough-approach-covid-19-hurting-citizens/ ). There were roadblocks manned by Military Police, the Army and Local defense personnel to ensure compliance. The measures were so strict to the extent of shooting some noncompliant citizens. Some of those acts are reflected in the BBC news article titled “Uganda - where security forces may be more deadly than coronavirus” (see https://www.bbc.com/news/world-africa-53450850 )

The approach of Rawson et al. 2020 in which the population was split into a quarantine and a non-quarantined group differing by the rate of virus transmission and the groups being connected by the modeled release strategies from lockdown seems to be one of the most suitable approaches for the countries in which lockdown restrictions were less strict.

We have added a justification for our choice of lockdown modeling in the discussion on page 18 in lines 380-386 as “On lockdown modeling, we assumed that a percentage of the susceptible population was isolated by the lockdown restrictions and hence impossible to become infected. This assumption is justified by the strictness in enforcing these measures in Uganda. However, for relatively lax implementation of the measures, alternative approaches such as that involving splitting the population into the quarantine and non-quarantine groups as done by [Rawson et al. 2020] could be more suitable.”

This is all excellent contextual information that could be included in a brief paragraph in the introduction to help stress the importance of this work.

This compliment is highly appreciated and following your suggestion, we have introduced the following sentence in the introduction on page 5 in lines 100-103 “…Additionally, although there are other approaches to modeling lockdown (e.g. [22]), our approach of having a given percentage of the susceptible population being totally unavailable to mingle was motivated by Uganda’s strictness in enforcing lockdown measures….”

When the model is introduced (from L102), it is very difficult to interpret the model initially, needing to jump between table 1, figure 1, equations (1)-(5), and brief paragraphs. I would instead urge presenting each equation, one-by-one, and explaining specifically each term (and parameter) in each equation. This may seem excessive, but makes interpreting a model considerably easier.

The flow of content in this section has been improved up by numbering and then describing each equation fully in the text right below where all the equations appear. The content is capture on page 9 in lines 179-200 as “In Equation (3.1), the term (1-(A+E))ΛN represents the number of arriving susceptible individuals per day, ΒBS(QI_A+I_S+GH)/N represents the number of susceptible individuals that become latently infected per day, ΤR represents the number of previously immune individuals that lose their disease-induced immunity to become susceptible per day and ΛS represents the number of departing susceptible individuals per day. In Equation (3.2), the term EΛN represents the number of arriving latently infected individuals per day while ΡE represents the number of latently infected individuals that progress to become infectious per day. In Equation (3.3), the term AΛN represents the number of arriving asymptomatically infectious individuals per day, RΡE represents the number of latently infected individuals that become asymptomatically infectious per day and Ω_A I_A represents the number of asymptomatically infectious individuals that are hospitalized per day. In Equation (3.4), the term (1-(C+R))ΡE represents the number of latently infected individuals that become symptomatically infectious per day, Σ_S I_S represents the number of symptomatically infectious individuals that succumb to the disease before being hospitalized per day and Ω_S I_S represents the number of symptomatically infectious individuals that are hospitalized per day. In Equation (3.5), the term CΡE represents the number of latently infected individuals that were traced and isolated prior to being hospitalized upon testing positive per day, Ω_A I_A represents the number of asymptomatically infectious individuals that are hospitalized per day, Ω_S I_S represents the number of symptomatically infectious individuals that are hospitalized per day, Σ_H H represents the number of hospitalized individuals that succumb to the disease per day while ΑH represents the number of hospitalized individuals that recover from the disease per day. Lastly, in Equation (3.6), the term ΛR represents the number of departing recovered individuals per day and the other two terms are described above.”

The model is now far easier to read and interpret to new readers, many thanks for these changes.

Thank you for all your efforts and devotion towards improving this manuscript.

…

Minor Comments:

L51/L52 – “measures reduced the susceptible population to about 10%”, this is important, is there a citation for this?

This was mentioned in press briefings by the Ministry of Health and other non-documented outlets. However, it can be approximated based on the work demographic data of Uganda (Source: Uganda Bureau of Statistics) where 80% of Ugandan’s workforce is employed in the non-formal sector and hence were less likely to have been categorized as “Essential workers” as per the definition used during lockdown. Only a few market vendors (who were not even allowed to return home to their families) as well as essential, health service providers and essential cargo transporters were allowed to operate. Also of the 43 million Ugandans, 15 million are school going and hence congregate in schools yet all schools were closed during lockdown.

Can you cite this particular press-briefing? Or mention these specific points somewhere in the manuscript to help explain this point. 

This has been done.

L129 – Does this mean that asymptomatic individuals are hospitalised? I assume this is due to the test-and-trace, this needs to be changed.

Yes. Everyone that tested positive- symptomatic or not- had to be hospitalized to prevent them from spreading the disease in the community. This went on until recently (as recent as early November 2020) when the hospitals got overwhelmed.

Again, this makes a lot more sense now that I understand the Uganda-specific protocol! Thank you for explaining it.

Our sincere apologies for the initial lack of clarity.

SUMMARY

The revisions made address almost all of my points very well. There is only one point that needs addressing before I can finally recommend for publication, and that is regarding the model parameter values again. This work will be extremely useful in the coming years I believe for investigating the impact of different mitigation strategies, and as such it is important to make sure that these parameter values are well cited and explained for future readers to understand the measures unique to Uganda.

The reviewer’s foresightedness on the aspect of future usability of the model is greatly appreciated and the changes have been implemented as suggested.

Essentially I would just want to see some more information for those that cite reference [4]. Naturally, some of these parameters will be deeply uncertain, and so “back-of-the-envelope” estimates will need to be used. There is nothing wrong with this, however I would want to see exactly how these values were reached. As this could involve a lot of text, I would perhaps recommend moving these calculations to a separate appendix or supplementary info page. 

Back-of-the-envelope estimates could not entirely be avoided as the pathogen is novel and document archiving isn’t all that the best. We have amended some of the cited literature to reflect “Estimated” or “Assumed” and for clarity on some of the parameters, we opted to add footnotes detailing the approached used to infer their values.

Also, where specific press-releases or ministerial briefings are used, these specific briefings should be cited, such as the ones that the authors linked to me in the above comment regarding influx from other countries / lockdown amounts. In short, a lot of the justification given to me in responses by the authors should be included for all readers!

Whenever available, the relevant briefs have now been cited as recommended.

Really the main sticking point here is still the mortality rate. These estimates were based on old data that is clearly now not an accurate representation of the true mortality rate to my eyes. Indeed, based on today’s case data the total mortality rate is 0.007 in Uganda for confirmed cases. The current parameter value used is likely to really affect the results shown, which is important given some of the findings regarding hospital capacity. I think these really need to be re-run with an improved mortality rate, unless my understanding is incorrect?

As advised and given the prevailing disease dynamics, we have revised the mortality rate to 0.008 and maintained a 10% reduction factor for the hospitalized.

Once this is addressed I will be happy to recommend for publication. 

A minor point, I think it would be extremely helpful for reviewers if a paragraph was included in the Introduction that outlines the specific, extreme, protocols used in Uganda. This motivates the work extremely well, showing why bespoke modelling frameworks are needed. 

These have been introduced as detailed above. Very much appreciated!

Finally,

L295 – “Even more worrying is the fact that the endemic level would higher at bigger levels of susceptibility” – I think there’s a typo here, I’m a bit confused about what is trying to be said with this sentence?

This has been amended on page 16 line 315-316 as “… Even more worrying is the fact that the prevalence level would be higher at higher levels of susceptible population….”

---

## [Editor Report · Decision Letter 3]

8 Feb 2021

Mathematical modeling of COVID-19 transmission dynamics in Uganda: Implications of Complacency and Early Easing of Lockdown

PONE-D-20-24477R3

Dear Dr. Ssematimba,

We’re pleased to inform you that your manuscript has been judged scientifically suitable for publication and will be formally accepted for publication once it meets all outstanding technical requirements.

Kind regards,

Siew Ann Cheong, Ph.D.

Academic Editor

PLOS ONE
---

## [Editor Report · Acceptance letter]

10 Feb 2021

PONE-D-20-24477R3 

Mathematical modeling of COVID-19 transmission dynamics in Uganda: implications of complacency and early easing of lockdown 

Dear Dr. Ssematimba:

I'm pleased to inform you that your manuscript has been deemed suitable for publication in PLOS ONE. Congratulations! Your manuscript is now with our production department. 

Kind regards, 

on behalf of

Dr. Siew Ann Cheong 

Academic Editor

PLOS ONE